# FedSCM: Federated Domain Adaptation via Sparse Consensus Matching

## Abstract

We address the problem of client selection for source-free domain adaptation in heterogeneous federated learning (FL). In this setting, a central server possesses an unlabeled target-domain dataset and aims to learn a model by leveraging locally trained models from a large pool of $K$ non-IID clients. Crucially, only a small subset of clients have data distributions that meaningfully align with the server's target domain, making effective client selection essential. However, due to strict privacy constraints, the server cannot access raw client data, client-side statistics, or labels for its own dataset—it can only evaluate the initially-trained client models on unlabeled target samples. To tackle this challenge, we propose Federated Sparse Consensus Matching (FedSCM), a principled, optimization-based method for label-free and data-free client selection. FedSCM selects clients whose predictions are both confident and mutually consistent, by solving an entropy-regularized sparse optimization problem over client weights. We prove that FedSCM always yields a sparse solution, and under a novel Dirichlet-based expertise model, it identifies the correct subset of relevant clients with high probability, provided $n \geq \Omega(K)$ target samples. We further establish local and global convergence guarantees under mild conditions. Extensive experiments on CIFAR-10, CIFAR-100, and SVHN demonstrate that FedSCM consistently outperforms existing approaches to federated domain adaptation, while significantly reducing both communication and computation overhead. Our framework offers a general and theoretically grounded approach to selective model aggregation under extreme data heterogeneity and limited supervision.

## 1 Introduction

Federated Learning (FL) has emerged as a transformative paradigm in distributed machine learning, enabling multiple clients to collaboratively train a shared global model without exposing their private local data. By exchanging only model updates rather than raw data, FL preserves privacy while leveraging the distributed intelligence of heterogeneous devices. This privacy-preserving approach has seen wide adoption in domains such as healthcare, finance, and mobile computing, where data confidentiality is critical (McMahan et al., 2017; Ye et al., 2023; Huang et al., 2023a). Despite its potential, FL faces key challenges, particularly in large-scale deployments involving thousands or even millions of clients. At such scales, the computational load and communication overhead associated with iterative training become significant. Efficient client selection and communication strategies are thus essential for scalability (Huba et al., 2022).

Another major challenge arises from the *statistical heterogeneity* of client data. In practice, data across clients are typically non-independent or non-identically distributed (non-IID), which complicates model aggregation and convergence (Gao et al., 2022). This issue is especially pronounced in *Federated Domain Adaptation* (FDA), where the server aims to learn a model that performs well on an *unlabeled* target distribution that differs from the clients' data domains. FDA assumes that while the server has access to unlabeled target-domain data, the client data (source domains) should be sufficiently related to this target domain. If the divergence is too large, the adapted model's performance deteriorates sharply. Thus, ensuring that the source domains are not excessively dissimilar is crucial for the success of FDA, necessitating a principled mechanism to control the level of heterogeneity (Jiang et al., 2024).

A particularly challenging scenario arises when only a small subset of clients hold data distributions that are meaningfully aligned with the server's target domain. For instance, in social network analysis, most users' data may be irrelevant or even detrimental to the server's objective, introducing noise or bias during aggregation. Only a sparse subset of clients may contribute positively to model adaptation. Identifying and leveraging this subset is therefore critical. This motivates a key problem in FL—and in FDA in particular, which is also the focus of this paper: how to effectively eliminate irrelevant clients and retain only those with useful, target-aligned data. Doing so is essential to mitigate inefficiencies and performance degradation, and to enable a reliable and cost-effective post-selection federated training.

A central obstacle in addressing the above challenge is the stringent privacy constraints inherent in many FL settings, which prevent the server from accessing any direct statistics, raw samples or even labels from client datasets. Clients share only privatized initial models or model updates, and the server typically lacks ground-truth labels for its own data, making it difficult to evaluate or select suitable client models. Consequently, the server cannot assess the extent of potential *covariate* or *label shifts* in client data. This absence of supervision and metadata severely limits the applicability of existing client selection and domain adaptation methods, many of which rely on access to such auxiliary information (Zhang et al., 2023).

Despite the importance of client selection and domain adaptation in heterogeneous FL scenarios, this area remains under-explored. Most prior work emphasizes general optimization improvements or heuristic sampling strategies, with limited focus on principled methods that jointly address (i) the sparsity of relevant clients and (ii) the lack of accessible client statistics. Existing approaches can be grouped into three broad categories. First, federated domain generalization methods aim for robustness across client domains without leveraging the target distribution, often resulting in suboptimal adaptation accuracy. Many of these methods also lack scalability or do not address the issue of client sparsity (Zhang et al., 2023; Zhu et al., 2023), and often lack strong theoretical guarantees (Duan et al., 2021; Jeong & Hwang, 2022). Second, source-free domain adaptation methods—while theoretically aligned with FL's constraints—often violate privacy assumptions by requiring access to centralized or shared source data. They also tend to overlook the challenge of sparse client identification and often lack solid theoretical foundations (Feng et al., 2021). Finally, existing FDA methods frequently suffer from poor scalability and lack effective mechanisms for identifying and utilizing the most informative clients under strict privacy constraints (Ghannou & Bennani, 2024; Liu et al., 2023; Niu et al., 2023; Zhang et al., 2025; Feng et al., 2024; Qin et al., 2022b; Jiang et al., 2024; Yi et al., 2023a; Montesuma et al., 2024).

**Our Problem:** Motivated by the above facts, we now formalize our problem. Consider a federated learning system with $K$ clients. The central server has access to an unlabeled dataset $\mathcal{D} = \{\boldsymbol{X}_1, \ldots, \boldsymbol{X}_n\}$, comprising independent samples drawn from an unknown distribution $P$. The server lacks label information for $\mathcal{D}$. Each client $i \in \{1, \ldots, K\}$ possesses a local dataset sampled from its own private distribution $P_i$, which may significantly differ from both $P$ and the distributions of other clients. Clients train models/hypotheses $h_i$ locally—e.g., using neural networks—and send them to the server, resulting in a set of $K$ client models $\{h_1, \ldots, h_K\}$.

Given this setup, the server aims to effectively label its own unlabeled dataset $\mathcal{D}$. A naive approach would be to start a generic FDA procedure with all clients involved. However, this is problematic due to: (i) high computational and communication overhead, and (ii) distributional mismatch—since many client distributions $P_i$ may diverge significantly from $P$, the aggregated model may perform poorly on the server's data, defeating the purpose of domain adaptation.

A more targeted strategy would involve estimating the alignment between each client distribution $P_i$ and the server's target distribution $P$, and selecting only those clients whose data are more compatible. However, as discussed before, two key obstacles arise: i) the server cannot access any client data or statistics—only the model $h_i$ from client $i$ is available, and (ii) the server's dataset is unlabeled, preventing direct evaluation of the client models' performance on it.

In contrast to Byzantine-robust heterogeneous FL (Cao et al., 2020; Huang et al., 2024), which focuses on filtering out a small subset of malicious clients, our setting contains *no* adversaries. Instead, the challenge arises because only a *few* clients are aligned with the server's distribution $P$, while most clients are benign

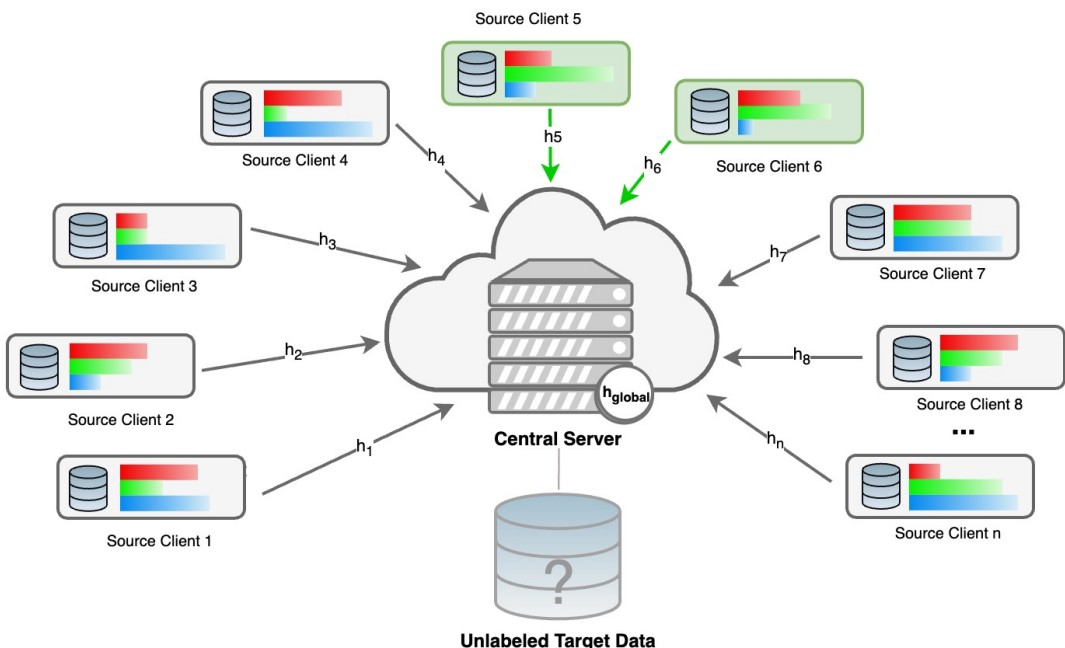

Figure 1: Overview of our federated learning setup: multiple clients each possess labeled datasets with diverse data distributions, while the server holds unlabeled data. Due to privacy constraints, clients only send trained model weights to the server, which then identifies relevant clients with data distributions aligned to the target. The system communicates selectively with these relevant clients to collaboratively train a robust base model for the target task.

yet distributionally unrelated. Consequently, the problem is not adversarial defense but identifying relevant clients under extreme heterogeneity and unlabeled data.

Figure 1 illustrates our federated learning problem setup, highlighting the distinct data distributions across clients and the server's role in managing unlabeled data.

## 1.1 Our Contribution

In this paper, we propose **Federated Sparse Consensus Matching (FedSCM)**, an optimization-driven framework to address the above challenge. FedSCM uses an entropy-based objective to identify a sparse subset of clients whose data distributions are most aligned with the server's unlabeled target domain without requiring access to client-side statistics or labels from the server's dataset. While prior works in heterogeneous FL use entropy primarily to identify confident clients (Huang et al., 2024; Park et al., 2021), FedSCM goes beyond confidence by also incorporating mutual consistency across client predictions and enforcing sparsity in client selection, enabling more robust and minimal subset selection. The core idea behind FedSCM is twofold: (i) Client models $h_i$ trained on distributions that are misaligned with the target domain $P$ tend to produce less confident predictions (i.e., logits close to uniform), whereas models trained on more relevant distributions tend to be more confident. (ii) Even when confidence is not a reliable signal, models that are well aligned with the target domain tend to make *similar* predictions across the server's unlabeled data, whereas unrelated models exhibit low consensus. FedSCM formalizes client selection as a constrained entropy-minimization problem, encouraging sparsity in aggregation weights while maximizing agreement among client predictions. Notably, the optimized weights are not meaningful importance scores. Only their zero or nonzero status matters: clients with nonzero weights are selected, and the magnitude of the weight has no semantic role. Unlike heuristic methods—which rely on ad hoc rules or similarity clustering

with limited guarantees—FedSCM jointly optimizes client weights to maximize both confidence and mutual consistency, while explicitly limiting each client's influence through a sparsity constraint. By evaluating all client models on the server's samples, FedSCM identifies a sparse subset of clients that are both confident and mutually consistent in their predictions—achieving data-free, label-free client selection for federated domain adaptation. In FedSCM, clients transmit their models to the server instead of raw data, which inherently provides a degree of privacy since the server never directly accesses the underlying client data. However, privacy is not a primary focus or challenge addressed in this work. If stronger privacy guarantees are required, model-level techniques such as differential privacy (Geyer et al., 2017) can be applied prior to transmission, and these approaches can be seamlessly integrated into the FedSCM framework. From the theoretical perspective, we establish the following guarantees for FedSCM:

**Sparsity Guarantees:** FedSCM solves a continuous and non-combinatorial optimization problem. However, in Section 4.1, we prove in Theorem 3 that it always selects a sparse subset of clients with a user-defined sparsity level—that is, the exact number of clients to select.

**Accuracy Guarantees:** In Section 4.2, we provide a novel theoretical foundation for data-free, label-free, and heterogeneous settings in FDA. We introduce an "expertise" model in Definition 4, wherein each client's classifier behavior on target-domain data is modeled as a convex combination of the true (unknown) label distribution and a Dirichlet-distributed noise. Later, we empirically validate this model on real-world datasets in Section 6.3.5. Under this model, we show that if a small subset of clients possesses sufficiently high expertise while others have low expertise levels, and the server's sample size $n$ is at least $\Omega(K)$, then FedSCM recovers the correct subset of relevant clients with high probability. See Theorem 7 for the formal statement.

**Optimization and Convergence Guarantees:** FedSCM involves a continuous, non-convex optimization problem. We establish two main convergence results: First, Theorem 9 shows that standard nonlinear programming solvers (e.g., SLSQP) always converge to at least a local minimizer. Empirical evaluations in Section 6 show that the obtained local minimizers consistently happen to be highly relevant solutions. Second, Theorem 10 proves a stronger result: under the expertise model of Section 4.2, and under similar assumptions to Theorem 7 on sample complexity, a simple Projected Gradient Descent (PGD) algorithm can achieve the *global* optimum if equipped with an exact line search method.

**Experimental Validation:** We also conduct comprehensive experiments on benchmark datasets constructed from CIFAR-10, CIFAR-100, and SVHN, simulating a range of distribution shifts: We model **Label and Quantity Shifts** via Dirichlet sampling, modeling heterogeneous class proportions and data imbalances. Next, we consider **Covariate Shifts** via affine noise augmentations, simulating domain mismatch in input distributions. FedSCM consistently outperforms state-of-the-art baselines in federated learning and domain adaptation, achieving higher accuracy on the target domain across all benchmarks. Moreover, by selecting only a small, relevant subset of clients, it significantly reduces communication and computation costs in post-selection FL or FDA procedures compared to full participation. We also empirically demonstrate its robustness under extreme distribution shifts—in both label proportions and input covariates. Detailed results are presented in Section 6.

It is worth noting that the privacy preservation offered by FedSCM is informal in nature: by operating solely on client model parameters rather than raw data, the method limits one channel of information exposure inherent in standard FL communication. This should not be interpreted as a formal privacy guarantee — in particular, it does not protect against attacks such as membership inference or model inversion. Stronger guarantees, such as those provided by differential privacy, are orthogonal to and compatible with the proposed framework, but are left to future work.

The remainder of the paper is organized as follows: Section 2 reviews related work. Section 3 introduces notation. Sections 4 and 5 present our method and theoretical results. Section 6 provides empirical validation, and Section 7 concludes the paper.

## 2 Background

Our study addresses domain adaptation in scalable FL, specifically when only a small subset of clients share meaningfully-aligned distributions with the target domain—a challenging and underexplored problem. To the best of our knowledge, no work directly tackles this issue. However, insights can be drawn from multiple related areas, including multi-source-free domain adaptation (MSFDA), federated domain generalization (FedDG), and federated domain adaptation (FDA).

### 2.1 Multi-Source-Free Domain Adaptation (MSFDA)

MSFDA builds on Source-Free Domain Adaptation (SFDA), extending its principles to scenarios with multiple source domains. Techniques like SHOT (Liang et al., 2020), USD (Jahan & Savakis, 2024), and CPD (Zhou et al., 2024) refine models using target domain data through pseudo-labeling, self-training, and class prototype alignment. MSFDA methods such as KD3A (Feng et al., 2021) and FMDA-OT (Ghannou & Bennani, 2024) tackle challenges such as irrelevant source weighting and privacy-preserving collaborative learning. MSFDA can be centralized, where source knowledge is integrated via an aggregator, or decentralized, facing issues such as bandwidth constraints and aggregation complexity. These approaches typically focus on alignment strategies like latent space transformation and intermediate domain generation, as well as matching strategies involving domain/sample weighting (Zhao et al., 2024). The above methods do not assume (nor take advantage of) the inherent sparsity of related clients.

### 2.2 Federated Domain Generalization (FedDG)

This category of methods aim to build global models that generalize across non-iid client distributions, enhancing robustness and scalability in FL. Data-level approaches such as augmentation and anonymization improve dataset consistency (Li et al., 2021; Yoon et al., 2021), while privacy-preserving techniques (Xu et al., 2022; Huang et al., 2022) further strengthen robustness. Model-level methods emphasize global model adaptability, with personalized federated learning (T. Dinh et al., 2020; Huang et al., 2023b) using partial model sharing and elastic configurations to address non-iid data. Server-level methods like FedGroup (Duan et al., 2021) and IFCA (Ghosh et al., 2020) optimize client clustering and aggregation. Domain Generalization methods such as FedDG (Zhang et al., 2023) dynamically adjust aggregation weights, while pFedVEM (Zhu et al., 2023) uses variational expectation maximization to address label skew. Hybrid methods like Factorized-FL (Jeong & Hwang, 2022) combine server and model-level strategies, reducing communication costs and improving compatibility. However, the theoretical foundations for weighted aggregation and clustering require further development.

### 2.3 Federated Domain Adaptation (FDA)

FDA focuses on transferring knowledge from multiple source domains to a target domain while preserving privacy. FOSDA (Qin et al., 2022a) prioritizes uncertain user contributions for global model construction, while RF-TCA (Feng et al., 2024) accelerates adaptation using transfer component analysis. Contrastive learning approaches using transformers (Yi et al., 2023b) enhance feature extraction and domain alignment. Other FDA frameworks such as FedDAD (Li et al., 2022) address privacy in object detection, while Co-MDA (Wang et al., 2021) improves adaptation through dynamic source weighting. Some methods like SDEA (Huang et al., 2024) use entropy of outputs to select relevant sources, but relying on client outputs is not robust due to noisy or miscalibrated predictions. Innovations like Federated Gradient Projection (FedGP) (Jiang et al., 2024) filter negative gradient components, and Multi-prototype Federated Fine-tuning (MPFT) (Zhang et al., 2025) uses domain-specific prototypes for improved knowledge transfer. Techniques like Mutually Collaborative Knowledge Distillation (MCKD) (Niu et al., 2023) and Federated Dataset Dictionary Learning (FedDaDiL) (Montesuma et al., 2024) promote domain-invariant representations and robust adaptation. However, FDA methods still face challenges when target data is server-confined and unlabeled, which limits real-world applicability where privacy constraints and limited data movement are significant concerns.

Despite these advancements, current MSFDA, FedDG, and FDA approaches face key limitations, including challenges in effectively weighting diverse sources, addressing iterative refinement in decentralized setups, and ensuring practical deployment under stringent privacy constraints. This motivates the need for innovative methods that enable more efficient and scalable domain adaptation in federated learning.

## 3 Notation

For $n \in \mathbb{N}$, we denote the set $\{1, 2, \ldots, n\}$ by $[n]$. For any two sets $\mathcal{I}, \mathcal{I}' \subseteq [n]$, we define $\mathcal{I} - \mathcal{I}' \triangleq \{i \in \mathcal{I} \mid i \notin \mathcal{I}'\}$. For $C \geq 2$, the set $\Delta^{C-1}$ denotes the $(C-1)$-dimensional simplex, the set of all possible $C$-dimensional probability mass functions:

$$\Delta^{C-1} \triangleq \left\{ \boldsymbol{p} \in \mathbb{R}^C \mid \boldsymbol{p} \succeq \boldsymbol{0}, \; p_1 + p_2 + \ldots + p_C = 1 \right\}.$$

Let $\mathcal{X} \subseteq \mathbb{R}^d$ be a measurable feature space, and $\mathcal{Y}$ represent a set of discrete labels. For a multi-class classification task, we have $\boldsymbol{Y} = [C]$, where $C \geq 2$ is the number of classes. The data-generating distribution $P$ is a probability measure supported on $\mathcal{X} \times \mathcal{Y}$, i.e., the space of feature-label pairs. The marginal distributions are $P_{\mathcal{X}}$ over $\mathcal{X}$, $P_{\mathcal{Y}} \in \Delta^{C-1}$ over $\mathcal{Y}$, and the conditional distribution $P(\cdot \mid \boldsymbol{X}) \in \Delta^{C-1}$ for any $\boldsymbol{X} \in \mathcal{X}$. In the formal problem definition of Section 1, we have assumed $\boldsymbol{X}_1, \ldots, \boldsymbol{X}_n \in \mathcal{D}$ are independent samples of $P_{\mathcal{X}}$.

We define the *entropy* and *KL-divergence* between discrete probability distributions as follows:

**Definition 1** (Entropy and KL Divergence). Let $\boldsymbol{p}, \boldsymbol{q} \in \Delta^{C-1}$ be two discrete probability measures over $[C]$. The entropy of $\boldsymbol{p}$, denoted $\mathbb{H}(\boldsymbol{p})$, and the Kullback-Leibler (KL) divergence between $\boldsymbol{p}$ and $\boldsymbol{q}$, denoted by $\mathsf{KL}(\boldsymbol{p} \parallel \boldsymbol{q})$, are defined as:

$$\mathbb{H}(\boldsymbol{p}) = \sum_{i=1}^C p_i \log \frac{1}{p_i} \quad , \quad \mathsf{KL}(\boldsymbol{p} \parallel \boldsymbol{q}) = \sum_{i=1}^C p_i \log \frac{p_i}{q_i}.$$

The entropy satisfies $\mathbb{H}(\boldsymbol{p}) \leq \log C$, with equality if $\boldsymbol{p}$ is the uniform distribution over $[C]$. The KL divergence is infinite if $p_i > 0$ and $q_i = 0$ for some $i \in [C]$.

Let $\mathcal{H}$ be a class of hypotheses for multi-class classification problem, such as the class of deep neural networks with an arbitrary architecture. We make no assumption regarding $\mathcal{H}$ or its complexity. Therefore, each $h \in \mathcal{H}$ is a mapping from $\mathcal{X}$ to $\Delta^{C-1}$, i.e., a probability vector over all possible labels. In later sections of the paper, such as Section 4.2, we discuss the notion of a *random* model, by which we mean that the randomness arises from training the model on a randomly drawn local dataset of a client in the network.

## 4 Federated Sparse Consensus Matching

In this section, we present our proposed method to address the problem outlined in Section 1, and then establish several theoretical guarantees in the subsequent subsections, the end-to-end workflow of our pipeline is depicted in Figure 2. Assume we wish to assign a weight $\lambda_i$ to each client $i$, where $\lambda_i \in [0, 1]$ and $\lambda_1 + \ldots + \lambda_K = 1$ (i.e., $\boldsymbol{\lambda} \in \Delta^{K-1}$). We aim for more relevant clients to receive larger values of $\lambda_i$, while sufficiently irrelevant ones should receive $\lambda_i = 0$. That is, we seek a *sparse* solution for the weight vector $\boldsymbol{\lambda}$. It is worth noting that the weights $\lambda_i$ serve solely as selection flags. Any client with $\lambda_i > 0$ is considered selected, while $\lambda_i = 0$ indicates exclusion; the specific value of a nonzero $\lambda_i$ carries no additional meaning. In this regard, we propose the following estimator:

**Definition 2** (Sparse Consensus Matching). For a fixed hyper-parameter $\gamma \in (0, 1)$, consider the following estimator $\boldsymbol{\lambda}$:

$$\boldsymbol{\lambda}^* \triangleq \underset{\boldsymbol{\lambda} \in \Delta^{K-1}}{\arg\min} \frac{1}{|\mathcal{D}|} \sum_{\boldsymbol{X} \in \mathcal{D}} \mathbb{H}\left( \sum_{i=1}^K \lambda_i h_i(\boldsymbol{X}) \right)$$
$$\text{subject to} \quad \lambda_i \leq 1 - \gamma, \quad \forall i \in [K]. \tag{1}$$

The final selected clients correspond to those indices $i \in [K]$ with non-zero $\lambda_i^*$.

The program in Definition 2 addresses a continuous and constrained optimization problem, leveraging the collected models $h_1, \ldots, h_K$ from all clients and the unlabeled data points in the server-side dataset $\mathcal{D}$. Each $h_i(\boldsymbol{X})$, for $i \in [K]$ and an unlabeled data point $\boldsymbol{X} \in \mathcal{D}$, represents a probability distribution over the label set $\mathcal{Y} = \{1, 2, \ldots, C\}$. Consequently, the convex combination $\sum_i \lambda_i h_i(\boldsymbol{X})$ with $\boldsymbol{\lambda} \in \Delta^{K-1}$, also forms a valid probability distribution and its entropy, computed by $\mathbb{H}(\cdot)$, is well-defined. The core idea behind our approach is to choose those clients (assigning non-zero values to their corresponding $\lambda_i$ coefficients) that exhibit the following two important properties:

- **High Individual Confidence**: The estimator $\boldsymbol{\lambda}^*$ in Definition 2 favors models whose prediction vectors $h_i(\boldsymbol{X})$ exhibit low entropy when averaged over all $\boldsymbol{X} \in \mathcal{D}$. This encourages the selection of clients that label the data points in $\mathcal{D}$ with relatively high confidence.

- **Consensus-Aligned Predictions**: High individual confidence alone can lead to disastrous outcomes, as a model from a random client might confidently predict incorrect labels. To ensure effective collaboration among models, we also select clients whose prediction probabilities $h_i(\boldsymbol{X})$ are aligned. This alignment means that the predictions maintain low entropy even when combined as a convex combination through non-zero coefficients $\lambda_i$.

In other words, FedSCM formulates an optimization problem that seeks to identify the optimal subset of client models that minimize uncertainty (entropy) in the aggregated predictions. This ensures that only a limited number of the most reliable and consensus-aligned client models are selected, thereby enhancing both efficiency and performance. However, several challenges and questions remain to be addressed. First, how does the program in Definition 2 generate a sparse coefficient vector $\boldsymbol{\lambda}^*$, and how does the choice of the hyperparameter $\gamma$ influence the level of sparsity? Second, under what conditions or assumptions can theoretical guarantees be established for the optimization output? Specifically, how do the dataset size $n$ and the network size $K$ affect the accuracy of the solution in a non-asymptotic regime? Third, how can the optimization program be practically solved?

In the following three subsections, we address these questions in detail. Section 4.1 proves that the optimal coefficients $\boldsymbol{\lambda}^*$ contain at most $\lceil 1/(1-\gamma) \rceil$ non-zero entries, thereby providing a certificate of sparsity. Section 4.2 introduces a natural and novel data generation model under which FedSCM provably identifies relevant clients for the unlabeled dataset, provided a suitable sample complexity is satisfied. Our bounds are both tight and non-asymptotic with respect to $K$ and $n$, offering a certificate of accuracy. Finally, Section 4.3 presents insights into the convergence behavior of generic optimization methods applied to the proposed objective.

## 4.1 Guarantee of Sparsity

Motivated by the objectives of this work, it is crucial to demonstrate that the optimization problem defined in Definition 2 produces a sparse solution. Specifically, we aim to show that the optimal solution $\boldsymbol{\lambda}^*$ activates only a limited number of clients. The following theorem formalizes this property:

**Theorem 3** (Sparsity Guarantee). *Assume that the $nC \times K$ matrix of client predictions over $\mathcal{D}$, formed by concatenating the prediction vectors $h_k(\boldsymbol{X})$ over all $\boldsymbol{X} \in \mathcal{D}$ for each client $k$, is full column rank. Then the optimization problem in Definition 2 admits an optimal solution $\boldsymbol{\lambda}^*$ that is exactly $\lceil 1/(1-\gamma) \rceil$-sparse. Specifically, in such an optimal solution, some $\lambda_i$ are equal to $1-\gamma$, others are zero, and at most one $\lambda_i$ lies strictly between $0$ and $1-\gamma$.*

The proof is provided in Appendix A. Theorem 3 establishes that FedSCM inherently enforces sparsity in client selection. Most clients are either assigned the maximum weight of $1 - \gamma$ or excluded entirely with a weight of zero. Since FedSCM selects at most $\lceil 1/(1 - \gamma) \rceil$ out of $K$ clients for the subsequent federated fine-tuning stage, the communication and computation cost of all later training rounds is reduced by a factor of $K/\lceil 1/(1 - \gamma) \rceil$ relative to full participation. This sparsity not only enhances computational efficiency for future inference but also improves the interpretability of the consensus model. The proof leverages the strict concavity of the objective under the full-rank assumption, together with the polytope structure of the feasible set, to validate this property.

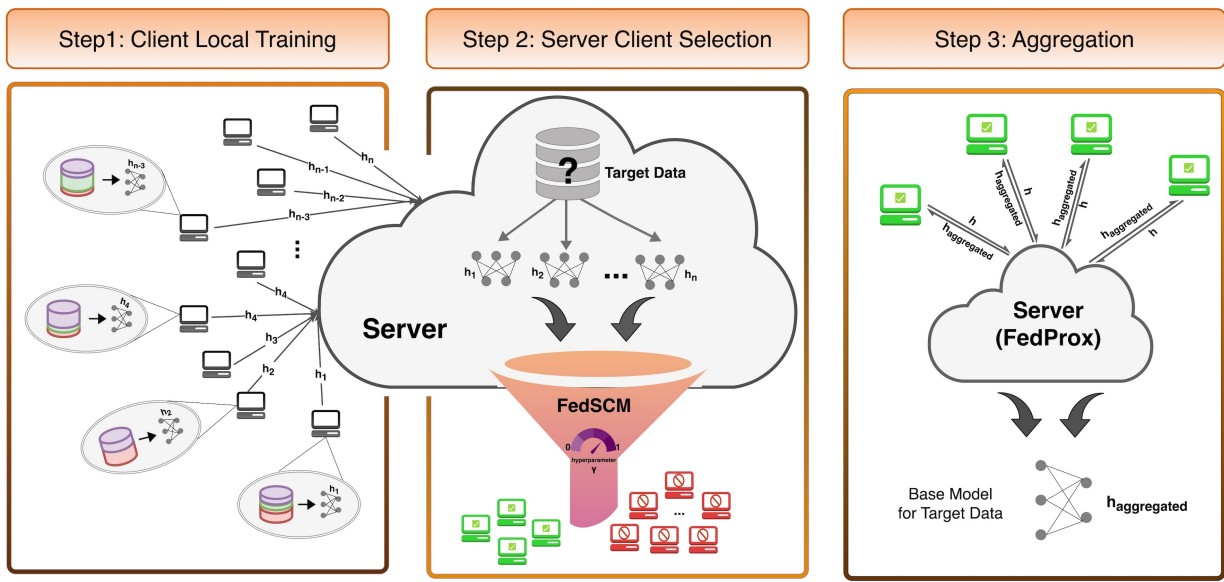

Figure 2: The proposed federated learning framework for domain adaptation Initially, each client possesses its own labeled dataset with distinct data distributions and trains a local model with a shared architecture. These client models then send their weights to the central server. Next, the server has access to target unlabeled data, which is input into each client model to obtain softmax outputs. The softmax predictions are then fed into the FedSCM module, which identifies a sparse group of the most relevant clients based on their outputs (controled by hyperparameter $\gamma$), effectively selecting models most suited to the target domain. In the final stage, the server collaborates iteratively with the selected clients using the FedProx algorithm to fine-tune a robust base model tailored for the target data distribution.

## 4.2   Guarantees on Client Selection Accuracy

In this section, we present a non-asymptotic analysis of the accuracy of client selection by FedSCM. Our goal is to show that the estimator reliably identifies the most competent clients, based on the relevance of their local data distributions to the server-side distribution $P$. However, such a guarantee is fundamentally unattainable without a theoretical model governing the generation of clients and their local data distributions.

**Dirichlet Noise Model**: We introduce a plausible and natural mathematical model for data and client generation. To keep the formulation both simple and realistic, we make as few assumptions as possible. Assume that a client with a local model $h$ is randomly selected from a large pool of clients in a federated network. Specifically, we formalize how to quantify whether a randomly chosen client possesses a model $h$ that is relevant to server-side data samples, which are drawn from an underlying latent distribution $P$. To enable this analysis, we introduce the concept of *model expertise*, which quantifies a client's reliability in predicting labels conditional on the data distribution.

**Definition 4** (Model Expertise)**.** We say that a (randomized) model $h\colon \mathcal{X} \to \Delta^{C-1}$ has an *expertise level* $\eta \in [0,1]$ on a latent true data-generating distribution $P$ if, for any feature vector $\boldsymbol{X} \in \mathcal{X}$, the predicted label distribution $h(\boldsymbol{X}) \in \Delta^{C-1}$ is given by the following statistical mixture:

$$Q \sim \mathrm{Dirichlet}\,(\alpha,\dots,\alpha) \tag{2}$$
$$h(\boldsymbol{X}) = \eta P(\cdot \mid \boldsymbol{X}) \; + \; (1-\eta)Q,$$

where $P(\cdot|\boldsymbol{X}) \in \Delta^{C-1}$ denotes the true conditional distribution of $y$ given $\boldsymbol{X}$, and $Q$ is a random distribution in $\Delta^{C-1}$ drawn from a Dirichlet$(\alpha,\dots,\alpha)$ distribution with some hyperparameter $\alpha > 0$.

The expertise level $\eta$ measures the extent to which the model's predictions align with the true conditional distribution $P(\cdot \mid \boldsymbol{X})$ versus relying on Dirichlet-based noise. It should be noted that while we typically focus

on deterministic models, such as deep neural networks, Definition 4 assumes a randomized model $h$. This is because the analysis considers $K$ clients sampled *randomly* from the network—i.e., we assume random client generation along with random local datasets. As a result, for any $\boldsymbol{X} \in \mathcal{X}$, the model output $h(\boldsymbol{X})$ becomes a random distribution over labels $\mathcal{Y} = [C]$.

*Remark* 5 (Modeling Philosophy). Definition 4 is not the result of identifying a narrow special case under which FedSCM happens to work. Rather, it is a mathematically tractable formulation of a phenomenon pervasive in federated learning under distribution shift: a classifier trained on a source domain, when evaluated on a target domain, behaves as if it partially tracks the true label distribution and partially produces uninformative or misleading outputs. More precisely, when a learner has access to the true conditional $P(\cdot|\boldsymbol{X})$, the optimal model is the Bayes classifier. Under distribution shift or limited data, however, the learned model deviates from this ideal. A natural way to model this deviation is through the mixture $h(\boldsymbol{X}) = \eta P(\cdot|\boldsymbol{X}) + (1 - \eta) Q$, where $Q$ represents the deviation from the Bayes classifier induced by the shift, and $\eta \in [0, 1]$ quantifies how much of the true signal is retained — what we call the client's *expertise*. When $\eta \to 1$ the client is essentially Bayes-optimal on the target; when $\eta \to 0$ its output is dominated by noise. The choice of a Dirichlet$(\alpha, \ldots, \alpha)$ distribution for $Q$, rather than the simpler uniform distribution, allows the noise component to be *confidently wrong* — placing most of its mass on an incorrect label — which makes the model strictly more general and more challenging than a uniform-noise baseline. We empirically validate this model on real-world datasets in Section 6.3.5.

**Main Result**: In a large federated network of size $K$ with heterogeneous client distributions, we assume the Dirichlet noise model from the previous subsection applies to client models with respect to an unknown data distribution $P$ at the server side. The server has access only to an empirical dataset of $n \geq 1$ *unlabeled* samples from $P_{\mathcal{X}}$ in $\mathcal{D}$. Additionally, we assume that while the true conditional $P(y \mid \boldsymbol{X})$ is unknown, the Bayes-optimal error for label estimation under $P$ is small. This assumption guarantees that low-error label estimation is theoretically possible; otherwise, the problem becomes fundamentally ill-posed and impossible to solve.

**Assumption 6** (Low Bayes Error for $P$). *Assume that for $\boldsymbol{X} \sim P_{\mathcal{X}}$, we have $\mathbb{H}(P(\cdot \mid \boldsymbol{X})) \leq b$ $P_{\mathcal{X}}$-almost surely, for some constant $b \geq 0$.*[1]

The following theorem ensures that, as long as (i) $n \geq \Omega(\log K)$, with constants depending only on the label size $C$ and the Dirichlet concentration parameter $\alpha > 0$, and (ii) a small subset of the $K$ clients have *relevant* distributions with respect to $P$ (i.e., high expertise $\eta$), while the rest are largely *irrelevant* (low expertise $\eta$), then FedSCM successfully identifies the latent cluster of relevant clients with high probability.

**Theorem 7** (Sample Complexity and Recoverability Guarantee). *Let $P$ be an unknown data-generating distribution with label size $C \geq 2$, satisfying Assumption 6 for some $b \geq 0$. The central server has access to $n \geq 1$ independent unlabeled samples from $P_{\mathcal{X}}$ and is connected to $K$ clients. Each client $i \in [K]$ returns a model $h_i$ with a latent expertise level $\eta_i \in [0, 1]$ with respect to $P$ under a Dirichlet noise model with concentration parameter $\alpha > 0$ (see Definition 4). Assume there exists a subset $\mathcal{I}^* \subset [K]$ of size $k^*$ (with both $\mathcal{I}^*$ and $k^*$ being unknown) such that, for some $\delta, \delta' \geq 0$, we have $\eta_i \geq 1 - \delta$ for $i \in \mathcal{I}^*$, and $\eta_i \leq \delta'$ for $i \notin \mathcal{I}^*$. For $\zeta \in (0, 1)$, assume*

$$\frac{b + \delta}{(1 - \delta')^2} \log\left(\frac{1}{b + \delta}\right) \leq \frac{\Psi_{C,\alpha}}{k^{*2}} \ , \ n \geq \max\left\{\Phi_{C,\alpha} \log\left(\frac{K}{\zeta}\right), \frac{K}{C}\right\} \tag{3}$$

*where $\Phi_{C,\alpha}$ and $\Psi_{C,\alpha}$ are constants depending only on $C$ and $\alpha$. Then, there exists $\gamma \in [0, 1]$ such that, with probability at least $1 - \zeta$, the minimizer of the FedSCM objective in Definition 2 with threshold parameter $\gamma$ exactly recovers $\mathcal{I}^*$, i.e., $\{i \mid \lambda_i^* > 0\} = \mathcal{I}^*$. Moreover, assuming $\delta, \delta'$, and $b$ are known, $\gamma$ can be determined.*

The proof, including the exact formulation of the constants, is provided in Appendix B. Theorem 7 establishes a strong theoretical foundation for the FedSCM method in the non-asymptotic regime. It guarantees that,

---

[1] For this assumption to be non-trivial, one must have $b < \log C$, since $\mathbb{H}(P(\cdot \mid \boldsymbol{X})) \leq \log C$ holds trivially for any distribution over $C$ classes. This almost-sure requirement can be relaxed in two equivalent ways: (i) requiring the bound to hold for at least a $(1 - \varrho)$-fraction of the samples in $\mathcal{D}$, for some small $\varrho \geq 0$; or (ii) replacing the almost-sure condition with the probabilistic requirement $\mathbb{P}_{\boldsymbol{X} \sim P_{\mathcal{X}}}(\mathbb{H}(P(\cdot \mid \boldsymbol{X})) \leq b) \geq 1 - \varrho$. Both relaxations lead to a modified sample-complexity condition in Theorem 7, and are analyzed in Appendix B.1.

as long as $n \geq \Omega(K)$ (sample complexity), and the parameters $(b, \delta, \delta')$ are sufficiently small, the true set of *relevant* clients is recoverable. Both conditions are well-motivated: If $n$ is too small, decisions generally become unreliable. On the other hand, a large $b$ implies a weak relationship between labels and features under $P$, making the problem fundamentally ill-posed. Similarly, large values of $\delta$ and $\delta'$ indicate poorly separated relevant and irrelevant clients, again rendering the problem infeasible.

**Scope and Limitations of the Entropy Heuristic.** The entropy-minimization objective underlying FedSCM is designed for *label shift* and *quantity shift*, where clients share the same labeling concept $P(y \mid \boldsymbol{X})$ and differ only in label proportions and dataset sizes. In this regime, predictive entropy on the server data is a reliable proxy for client relevance, and the guarantees of Theorems 7 and 10 are established precisely under this setting via the Dirichlet noise model of Definition 4.

Under *covariate shift*, where the input distributions differ across clients and the target, confidence may no longer correlate with relevance: a model trained far from the target distribution can produce highly confident but inaccurate predictions. Although the consensus-alignment effect in FedSCM partially mitigates this issue by penalizing confidently wrong clients that disagree with relevant ones, this correction is incomplete. Consequently, FedSCM is expected to be more effective under label and quantity shift than under severe covariate shift, consistent with our experiments.

More generally, this limitation reflects a no-free-lunch phenomenon: no selection rule based only on unlabeled target data and client predictions can be universally optimal across all forms of distribution shift. Any confidence-based criterion implicitly assumes that confidence correlates with relevance, an assumption that holds under label and quantity shift but not under arbitrary covariate shift.

### 4.3 Optimization and Convergence Guarantees

The FedSCM objective in Definition 2 is a non-convex function constrained to a convex set, rendering the overall optimization problem inherently non-convex. Consequently, standard convex optimization techniques cannot directly guarantee the global convergence of local, gradient-based methods such as Projected Gradient Descent (PGD). Nonetheless, as demonstrated in Section 6, our empirical results show that these algorithms—e.g., PGD and Sequential Least Squares Quadratic Programming (SLSQP)—often succeed in reaching the global minimum of the objective, thereby correctly identifying the set of relevant clients.

In this section, we present two theoretical results characterizing the global and local convergence properties of PGD and SLSQP in our setting. Prior to these results, we describe the implementation details of the PGD and SLSQP procedures in Algorithm 1.

**Proposition 8** (Gradient and Hessian of the FedSCM Objective). *The gradient $\boldsymbol{g}(\boldsymbol{\lambda}) \in \mathbb{R}^K$ (for any $\boldsymbol{\lambda} \in \Delta^{K-1}$) of objective in Definition 2 can be computed as:*

$$g_k(\boldsymbol{\lambda}) = \frac{1}{|\mathcal{D}|} \sum_{\boldsymbol{X} \in \mathcal{D}} \left[ \mathsf{KL}\left( h_k(\boldsymbol{X}) \,\middle\|\, \sum_{i=1}^{K} \lambda_i h_i(\boldsymbol{X}) \right) + \mathbb{H}\left( h_k(\boldsymbol{X}) \right) \right] - 1, \quad k \in [K]. \tag{4}$$

*Subsequently, the Hessian matrix $H(\boldsymbol{\lambda}) \in \mathbb{R}^{K \times K}$ can be computed as*

$$H_{j,k}(\boldsymbol{\lambda}) = -\frac{1}{|\mathcal{D}|} \sum_{\boldsymbol{X} \in \mathcal{D}} \sum_{i=1}^{C} h_j^{(i)}(\boldsymbol{X}) h_k^{(i)}(\boldsymbol{X}) \left( \sum_{\ell=1}^{K} \lambda_\ell h_\ell^{(i)}(\boldsymbol{X}) \right)^{-1}, \quad j, k \in [K]. \tag{5}$$

*As a result, the PGD and SLSQP algorithms for minimizing the objective for a given learning rate sequence $\{\alpha_t\}_{t \in \mathbb{N}}$ can be implemented according to Algorithm 1.*

Without imposing any assumptions on the data generation model, dataset size $n = |\mathcal{D}|$, or network size $K$, we show in Theorem 9 that the SLSQP algorithm is guaranteed to reach (at least) a local minimum of the objective function in Definition 2 that adheres to the sparsity structure described in Theorem 3.

**Theorem 9** (Local Convergence of SLSQP). *Assume the condition of Theorem 3 holds with certainty. Then any limit point of the sequence $\{\boldsymbol{\lambda}^{(t)}\}$ generated by the SLSQP method in Algorithm 1 satisfies the*

---

**Algorithm 1:** Projected Gradient Descent (PGD) and Sequential Least Squares Quadratic Programming (SLSQP) for FedSCM

---

**Input:** $K, \gamma, T$ (Number of PGD iterations), and $\{\alpha_t\}_{t \in [T]}$
$\mathcal{D} = \{\boldsymbol{X}_1, \boldsymbol{X}_2, \ldots, \boldsymbol{X}_n\}$: Unlabeled target dataset at the server
$\mathcal{H}_K = \{h_1, h_2, \ldots, h_K\}$: Locally-trained models from clients

**Initialize:** $\quad \lambda_k^{(0)} \leftarrow 1/K, \quad k \in [K].$

**for** $t \leftarrow 1$ **to** $T$ **do**
    **if** PGD algorithm is used **then**
        **for** $k \leftarrow 1$ **to** $K$ **do**
            $\lambda_k^{(t)} \leftarrow \max \left\{ \lambda_k^{(t-1)} - \alpha_t g_k \left( \boldsymbol{\lambda}^{(t-1)} \right), 0 \right\}$

    **if** SLSQP algorithm is used **then**
        $\boldsymbol{q}^{(t-1)} \leftarrow \underset{\boldsymbol{q} \in \mathbb{R}^K}{\arg \min} \ \frac{1}{2} \boldsymbol{q}^T H \left( \boldsymbol{\lambda}^{(t-1)} \right) \boldsymbol{q} + \boldsymbol{q}^T \boldsymbol{g} \left( \boldsymbol{\lambda}^{(t-1)} \right)$
        **for** $k \leftarrow 1$ **to** $K$ **do**
            $\lambda_k^{(t)} \leftarrow \max \left\{ \lambda_k^{(t-1)} + \alpha_t q_k^{(t-1)}, 0 \right\}$
        where $\boldsymbol{g}(\boldsymbol{\lambda})$ and $H(\boldsymbol{\lambda})$ are computed from Proposition 8

    **for** (at most) $\lceil 1/1 - \gamma \rceil$ times **do**
        $M \leftarrow \sum_{k \in [K]} \lambda_k^{(t)}$
        **for** $k \leftarrow 1$ **to** $K$ **do**
            $\lambda_k^{(t)} \leftarrow \min \left\{ \lambda_k^{(t)}/M, 1 - \gamma \right\}$

**Output:** $\boldsymbol{\lambda}^*_{\text{PGD or SLSQP}} \triangleq \boldsymbol{\lambda}^{(T)}$ (Final client weights)

---

*KKT conditions of the* FedSCM *objective in Definition 2. Moreover, every such KKT point is an extreme point of the feasible polytope and is at most $\lceil 1/(1 - \gamma) \rceil$-sparse. In particular, if the sequence converges, it converges to a sparse solution.*

The proof is provided in Appendix C. Per-iteration monotone decrease of the true objective is not guaranteed by Algorithm 1 as currently written, since each SLSQP step minimizes a local quadratic approximation of the objective without a step-size acceptance criterion; we refer the interested reader to standard references on SLSQP (Boggs & Tolle, 1995) for conditions under which a full monotone-decrease guarantee can be recovered.

Next, we consider the Dirichlet noise model introduced in Section 4.2. In Theorem 10, we show that under reasonable assumptions — such as a sufficiently large network size $K$ and server-side dataset size $n$ — the PGD method described in Algorithm 1 is, with high probability, guaranteed to converge to the *global* minimum of the FedSCM objective defined in Definition 2.

**Theorem 10** (Global Convergence of PGD under Dirichlet Noise Model)**.** *Consider the problem configuration of Theorem 7 with a Dirichlet noise model, and let Assumption 6 hold for $P$ with $b < \log C$. Assume there exists an unknown subset $\mathcal{I}^* \subset [K]$ such that $\eta_i \geq 1 - \delta$ for $i \in \mathcal{I}^*$ and $\eta_i < \delta'$ for $i \notin \mathcal{I}^*$, where $\delta + \delta' < 1$. Define the average expertise level as $\bar{\eta} \triangleq \frac{1}{K} \sum_{k=1}^{K} \eta_k$, and assume $\bar{\eta} > 0$. Furthermore, for $\zeta \in (0, 1)$, assume the following conditions hold:*

$$\frac{n}{\log(KC)} \geq \frac{\theta_1(b, C, \bar{\eta})}{(1 - (\delta + \delta'))^2} \log \frac{2}{\zeta}, \quad \text{and} \quad \frac{K}{\log(nC)} \geq \frac{\theta_2(b, C, \bar{\eta})}{(1 - (\delta + \delta'))^2} \log \frac{2}{\zeta}, \quad (6)$$

*where $\theta_1, \theta_2$ depend only on $b, C$, and $\bar{\eta}$, with at most degree-2 polynomial growth. Then, with probability at least $1 - \zeta$, the global minimum of the* FedSCM *objective is the unique KKT point of Definition 2, and any limit point of the PGD iterates in Algorithm 1 initialized at the uniform $\boldsymbol{\lambda}$ is this global minimum.*

The proof is provided in Appendix C. The conditions in equation 6 ensure that the dataset size $n$ and the number of clients $K$ are sufficiently large to guarantee that the signal from the most expert clients dominates

the gradient updates, smoothing the optimization landscape so that the global minimum is the unique stationary point. As a result, the limit-point guarantee of Theorem 9 — which holds without any assumption on the data generation model — specializes here to convergence to the global minimum specifically, rather than merely to an arbitrary sparse KKT point.

As discussed earlier, while the Dirichlet noise model may not perfectly reflect real-world settings, our empirical results in Section 6 indicate that SLSQP or PGD-like methods remain highly effective when the conditions of Theorem 10 are approximately satisfied, making them practical choices for large-scale optimization problems such as the one studied in this paper.

## 5   End-To-End Algorithm with Quantity Shift Mitigation

In this section, we propose an additional regularization term to mitigate the effect of *quantity shift*, which is prevalent in many federated learning (FL) scenarios. We then present the complete end-to-end pipeline of our method—including post-selection FL training—which will be used in the experiments of Section 6.

In many practical FL settings, a significant quantity shift arises when clients possess varying amounts of local data. Let $n_k$ denote the number of samples held by client $k \in [K]$. While the server typically does not have access to the exact values of $n_k$, in cases where this information is available, it can be used to prioritize clients with larger datasets. To account for this, we introduce a regularization term that penalizes client weights inversely proportional to their sample sizes. Specifically, we add a penalty of the form $\frac{\kappa}{n_k}$ to each $\lambda_k$, where $\kappa \geq 0$ is a tunable hyperparameter.

**Definition 11** (FedSCM with Quantity-Shift Regularization)**.** When the server has access to the local sample sizes $n_1, \ldots, n_K$, we modify the FedSCM objective as follows:

$$\boldsymbol{\lambda}^* = \underset{\boldsymbol{\lambda} \in \Delta^{K-1}}{\arg\min} \left[ \frac{1}{n} \sum_{i=1}^{n} \mathbb{H} \left( \sum_{k=1}^{K} \lambda_k h_k(\boldsymbol{X}_i) \right) + \kappa \sum_{k=1}^{K} \frac{\lambda_k}{n_k} \right], \tag{7}$$

subject to $\lambda_k \leq 1 - \gamma$ for all $k \in [K]$, where $\gamma \geq 0$ is a user-defined hyperparameter.

The regularization term $\kappa \sum_{k=1}^{K} \frac{\lambda_k}{n_k}$ incentivizes the selection of clients with larger datasets, enhancing model stability and reducing the influence of clients with small, potentially noisy local datasets.

We summarize the complete federated learning pipeline in Algorithm 2. The server collects local client models, applies the FedSCM objective to identify relevant clients, and subsequently performs federated training using only those clients with non-zero weights. For optimization, we employ FedProx, though alternative methods can also be integrated.

## 6   Experimental Results

This section presents an empirical evaluation of our proposed method across a range of real-world tasks, along with comparisons to state-of-the-art baselines. Due to space limitations, additional experiments are included in Appendix D. We first describe the datasets and experimental setup, then report and analyze results with a focus on client selection quality and final model accuracy, as determined by the end-to-end pipeline of Algorithm 2. Finally, we empirically validate the key assumptions underlying the expertise model and Dirichlet noise formulation, which support Theorems 7 and 10.

### 6.1   Datasets

We evaluate our approach using three standard real-world benchmarks—CIFAR-10 (Krizhevsky, 2009), CIFAR-100, and SVHN (Netzer et al., 2011). To assess the scalability of FedSCM, we simulate a federated learning environment with $K = 500$ clients. Each dataset is partitioned using various strategies to generate heterogeneous client populations, thereby emulating realistic non-IID federated settings. In such scenarios, clients exhibit diverse forms of distributional shift. In each experiment, the target distribution at the server-side is generated via the same procedure as other clients, and thus can be considered as the $(K + 1)$th client. We simulate three common types of heterogeneity widely studied in the literature:

---

**Algorithm 2:** (The Overall Pipeline) Client selection via FedSCM and subsequent federated model training via FedProx (Li et al., 2020)

---

**Input:** $K$, and $n_1, \ldots, n_K$; $\mathcal{D} = \{\boldsymbol{X}_1, \ldots, \boldsymbol{X}_n\}$; $\mathcal{H}_K = \{h_1, \ldots, h_K\}$
$\gamma, \kappa, \mu$ (Regularization hyper-parameters)$\geq 0$, $T$ (Number of FedProx iterations)
**Output:** $h^*$ (The final aggregated model for the target dataset)

**Step 1: Apply Client Models to Server-Side Unlabeled Data**
**for** $k \leftarrow 1$ **to** $K$ **do**
 Compute model outputs $\{h_k(\boldsymbol{X}_i) \mid \boldsymbol{X}_i \in \mathcal{D}\}$

**Step 2: Compute Client Weights via FedSCM**
Use Algorithm 1 to solve the optimization problem:

$$\boldsymbol{\lambda}^* \leftarrow \underset{\boldsymbol{\lambda} \in \Delta^{K-1}}{\arg\min} \left\{ \frac{1}{n} \sum_{i=1}^{n} \mathbb{H}\big(\boldsymbol{\lambda}^\top \mathbf{h}(\boldsymbol{X}_i)\big) + \kappa \, \boldsymbol{\lambda}^\top \mathbf{n}^{-1} \right\}$$

subject to $\lambda_k \leq 1 - \gamma$, $\forall k \in [K]$. Define the active set of clients: $\mathcal{I}^* = \{k \in [K] \mid \lambda_k^* > 0\}$

**Step 3: Initialize Global Model**
Initialize the global model $h^{(0)} \leftarrow \sum_{k \in \mathcal{I}^*} \lambda_k^* h_k$

**Step 4: Federated Optimization Using FedProx**
**for** $t \leftarrow 1$ **to** $T$ **do**
 **foreach** $k \in \mathcal{I}^*$ **do**
  Server sends $h^{(t-1)}$ to client $k$
  Let $\| \cdot \|_*$ denote any user-defined norm or similarity measures for models in the hypothesis set $\mathcal{H}$. Then, client $k$ updates the model via its local loss functional $\mathcal{L}_k(\cdot)$:

$$h_k^{(t)} \leftarrow \underset{h \in \mathcal{H}}{\arg\min} \ \mathcal{L}_k(h) + \frac{\mu}{2} \|h - h^{(t-1)}\|_*^2$$

  Client $k$ sends $h_k^{(t)}$ back to the server
 Server aggregates updated models: $h^{(t)} \leftarrow \sum_{k \in \mathcal{I}^*} \lambda_k^* h_k^{(t)}$

**Step 5: Output Final Model**
$h^* = h^{(T)}$

---

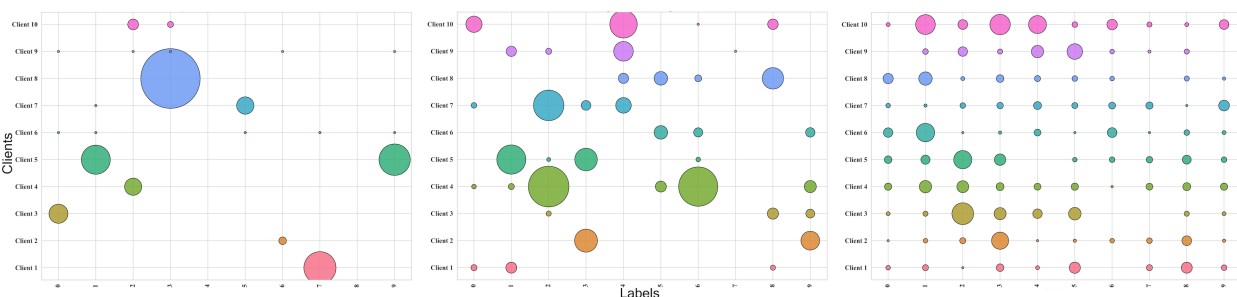

Figure 3: Class contribution per client in SVHN under different label shift levels (from left to right: $\alpha = 1$, 0.1, 0.01). Each circle represents a class, with size proportional to its representation in the client's local dataset.

- **Label shift:** Clients differ in class distributions. Specifically, any two clients $i, j \in [K]$ share the same conditional distribution $P_i(y|\boldsymbol{X}) = P_j(y|\boldsymbol{X})$ (i.e., the same *concept*), but have different marginal label distributions, i.e., $P_i(y) \neq P_j(y)$. Equivalently, by Bayes' rule, this means that the class-conditional feature distributions $P_i(\boldsymbol{X}|y)$ generally differ across clients as well, $P_i(\boldsymbol{X}|y) \neq P_j(\boldsymbol{X}|y)$, since they must adjust to keep the shared posterior $P(y|\boldsymbol{X})$ fixed while the label marginals $P_i(y)$ vary.

- **Covariate shift:** Clients have differing marginal feature distributions $P_i(\boldsymbol{X})$, often modeled via client-specific transformations applied to input data (Reisizadeh et al., 2020).

- **Quantity shift:** Clients vary in dataset size. Some clients have abundant local data, enabling more effective local training, while others operate with limited samples.

**Label and Quantity Shift.** To induce label shift, we follow prior work (Wang et al., 2020) and apply Dirichlet sampling to assign class distributions across clients, using concentration parameters $\alpha \in \{1, 0.1, 0.01\}$. Here, $\alpha = 1$ corresponds to nearly uniform class distributions, $\alpha = 0.1$ introduces moderate imbalance, and $\alpha = 0.01$ yields severe skew, with many clients receiving examples from only a few classes. This procedure simultaneously induces quantity shift, as class imbalance leads to varying numbers of local samples per client. Figure 3 illustrates these effects for CIFAR-10, where we have depicted class distribution for a randomly selected set of 10 clients. As shown, larger $\alpha$ results in more uniform distributions, while smaller values produce highly skewed class assignments.

**Covariate Shift.** Following (Reisizadeh et al., 2020), we simulate covariate shift by applying client-specific affine transformations to the input features. For each client $k \in [K]$, the transformed feature vector $\widetilde{\boldsymbol{X}}^{(k)}$ of the original image $\boldsymbol{X}^{(k)}$ is defined as: $\widetilde{\boldsymbol{X}}^{(k)} = (\boldsymbol{I} + \boldsymbol{\Lambda}^{(k)})\boldsymbol{X}^{(k)} + \boldsymbol{\delta}^{(k)}$, where matrix $\boldsymbol{\Lambda}^{(k)}$ and vector $\boldsymbol{\delta}^{(k)}$ are (entry-wise) drawn i.i.d. from zero-mean Gaussian distributions with standard deviation $\sigma = 0.05$. This results in diverse feature spaces across clients. Figure 4 shows sample images from clients under this transformation.

## 6.2 Experimental Setup

At the end of the client generation process described above, each client trains a separate local model—specifically, an artificial neural network—using only its own local dataset. All experiments use the VGG-11 architecture (Simonyan & Zisserman, 2015). Our objective is not to achieve state-of-the-art accuracy, but rather to isolate and evaluate the effects of client selection induced by FedSCM. Accordingly, we intentionally avoid more advanced architectures such as Vision Transformers. Each experiment is repeated three times with different random seeds, and we report the mean and standard deviation of the results. All experiments are conducted on a single NVIDIA RTX 4090 GPU.

## 6.3 Numerical Results

We now present our main experimental findings. We begin by verifying that FedSCM selects a sparse subset of clients in a controlled manner, guided by the parameter $1 - \gamma$, in accordance with Theorem 3. We then evaluate the "quality" of the selected clients using two key metrics:

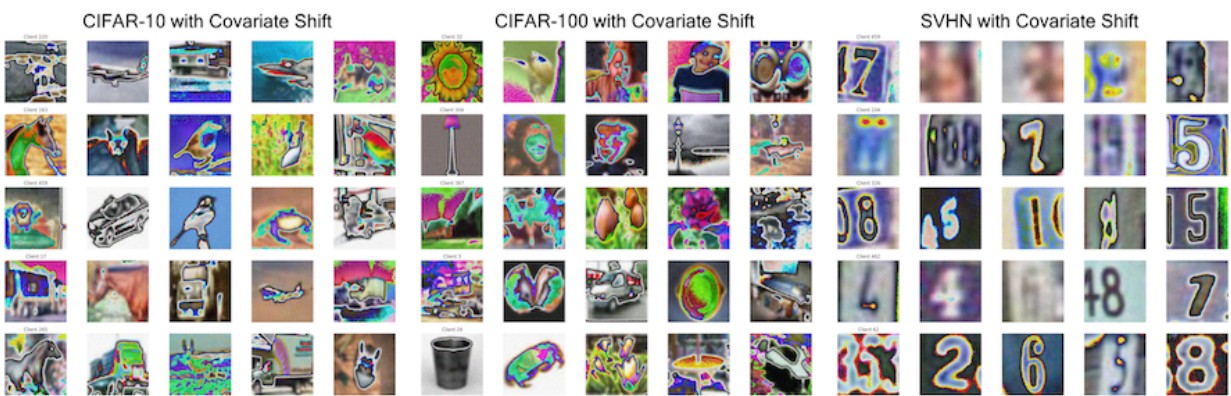

Figure 4: Sample images from five randomly selected clients per dataset (CIFAR-10, CIFAR-100, SVHN), each subjected to distinct affine transformations.

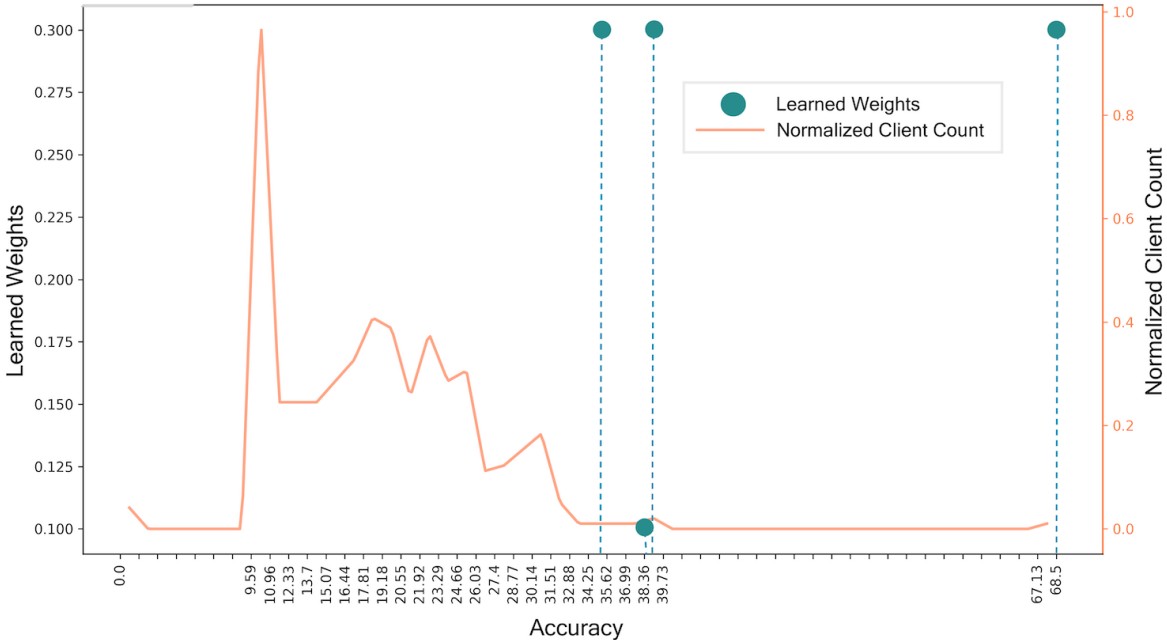

Figure 5: (**Green plot**) Client weights vs. accuracy on the target dataset. Horizontal axis: client accuracy. Vertical axis: learned selection weights. (**Orange plot**) Histogram of client accuracies. Note: vertical scales differ. This figure highlights (i) sparsity in selected clients and (ii) that all selected clients lie in the high-accuracy tail.

1. **Latent accuracy on server-side data:** Via having access to ground-truth labels on the server-side target dataset, we evaluate each selected client's local model. Higher accuracy on this validation set indicates a *better* client with more potentials for the downstream FL procedure.

2. **Final test accuracy after federated training:** We complete the client selection, and then proceed (according to Algorithm 2) for a post-selection FL. Hence, we assess the performance of the final global model on the server-side data. Higher final accuracy suggests that the selected clients collectively contributed to a well-generalized global model, validating the effectiveness of the selection process.

   To enable a fair comparison between different models, we restrict training to 1000 iterations. This ensures that the reported final test accuracies reflect differences arising from the model architecture and client selection, rather than improvements simply due to extended training.

We also examine the stability of the client selection mechanism across multiple runs and provide a comparative analysis against various baseline approaches.

### 6.3.1  Sparsity and Client Selection Quality

This subsection aims not to compare FedSCM with baselines but to demonstrate its behavior in practice. For brevity, we focus on scenarios involving label and quantity shift; results for covariate shift are presented later in Section 6.3.3, where we conduct comprehensive comparisons against state-of-the-art (SOTA) baselines. Our evaluation focuses on the client selection process using the CIFAR-10 dataset. Given the client weights $\lambda_i$ for $i \in [K]$, we identify the subset of selected clients as those with non-zero coefficients.

Figure 5 visualizes each client's accuracy on the target dataset (horizontal axis) against its corresponding selection weight (vertical axis). To aid interpretation, we include a histogram of client accuracies shown in orange. As expected, clients with higher target accuracy receive larger weights, indicating that FedSCM

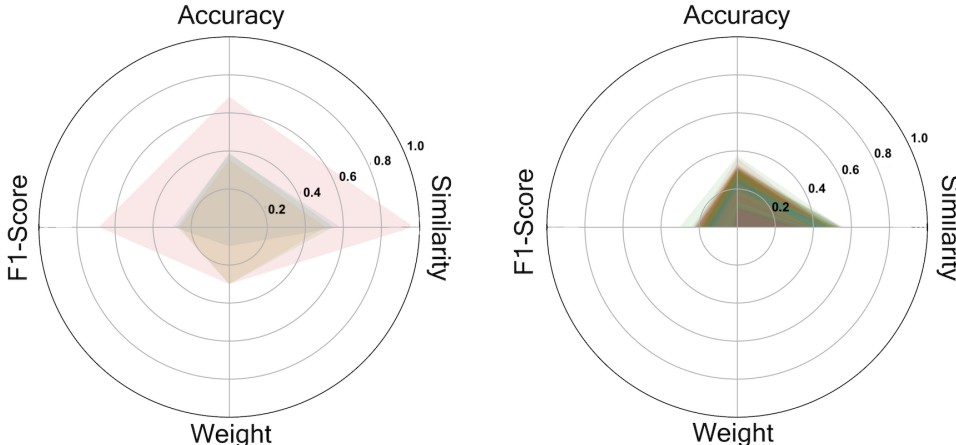

Figure 6: Radar chart for selected (left) vs. unselected (right) clients, showing accuracy, F1, and data similarity.

effectively prioritizes well-performing clients. Notably, all selected clients fall within the upper tail of the accuracy distribution, validating that FedSCM consistently identifies the most promising clients. The histogram further shows that most clients perform poorly on the target data—consistent with the presence of label shift—supporting the sparsity assumption that only a few clients possess data aligned with the target distribution.

The hyperparameter $\gamma$ was selected via cross-validation, with $\gamma = 0.7$ (resulting in the selection of 4 clients) used across all datasets. This setting strikes a balance between performance, accuracy, and computational cost. Results for additional datasets, as well as other values of $\alpha$ and $\gamma$, are available in Appendix D.

The radar charts in Figure 6 (left and right) compare the performance of selected (non-zero weight) and unselected (zero weight) clients under label shift on CIFAR-10 with $\alpha = 0.1$. Each radar axis represents a key metric: accuracy, F1 score, and similarity to the target distribution. Accuracy and F1 are evaluated using ground-truth labels from the target dataset. Similarity is computed as $1 - \mathrm{JSD}$, where JSD is the Jensen-Shannon divergence between a client's label distribution and the target label distribution. The results show that selected clients exhibit consistently higher accuracy and F1 scores and that their distributions are more similar to the target. This confirms that FedSCM effectively identifies clients whose data is best aligned with the target distribution. Additional results for other datasets and for different values of $\alpha$ and $\gamma$ are provided in Appendix D.

**Open Set:** We conducted additional experiments to show how client-side training strategies— implemented without compromising user privacy— can enhance server-side performance. Specifically, to address model overconfidence in underrepresented or unseen classes, we employed Open-Set learning techniques.

In imbalanced or limited-sample regimes, models may make overconfident predictions for a small subset of classes. Open-Set learning combats this by encouraging the model to recognize "unknown" or out-of-distribution samples. We used the ARPL (Adversarial Regularized Predictive Learning) method (Chen et al., 2021) within each client. ARPL encourages prediction uncertainty on unseen classes, leading to more balanced output distributions across all labels. Client models trained with ARPL were then uploaded to the server for the client selection and subsequent (limited)-federated training process. This approach led to more robust and stable outputs compared to standard training, albeit with increased computational demands at the client side. Figure 7 (left and right) illustrate performance gains in a 500-client setting.

### 6.3.2 Stability

Due to the stochastic nature of our optimization algorithms (e.g., SGD or SLSQP with random initializations), repeated runs of FedSCM may result in the selection of slightly different client subsets, even though

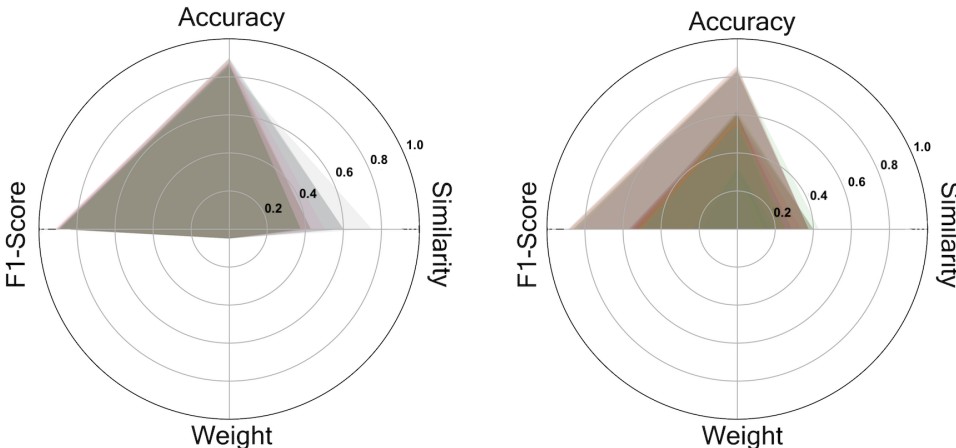

Figure 7: Comparison of selected vs. unselected clients via our method: (left) Selected clients with Open-Set learning (ARPL) for $K = 500$ clients, and (right) Unselected clients under the same Open-Set training.

the selected clients are consistently high quality. It should be noted that the number of selected clients remain constant since it is governed by the sparsity parameter $\gamma$. Recall that to improve selection stability, we propose an alternative formulation (Definition 11) that incorporates a size-based regularization term favoring clients with larger local datasets.

To compare the stability of the original and the modified objectives, respectively presented in Definitions 2 and 11, we executed both variants 50 times on the same dataset, under identical settings but with different random seeds. Each run produced a binary selection vector indicating which client samples were chosen. We then assessed the stability by comparing these selection vectors across all pairs of runs. Specifically, we compared the results using the following metrics: **Disagreement Ratio** which measures the proportion of samples that are selected in one run but not in another, **Hamming Distance** that counts the number of differing bits between two binary vectors, **Entropy** which quantifies the uncertainty in selection frequency across runs for each sample, and finally **Cosine Dissimilarity** (1 minus cosine similarity) which captures the angular dissimilarity between selection vectors.

For each metric, we computed the pairwise distances between all combinations of the 50 runs, and reported the average as a measure of overall stability. As shown in Table 1, the modified (regularized) objective consistently leads to lower variation across runs, confirming that it yields more stable selections.

### 6.3.3 Model Evaluation and Comparison

This section presents a comparative analysis of our method against state-of-the-art techniques for client selection in federated learning. Most of the baseline methods incorporate some form of client selection or weighting, aiming to adapt the final FL model to a target dataset, which is typically unlabeled. A detailed description of the compared methods is provided below:

- Source-Free Domain Adaptation methods such as K3DA (Feng et al., 2021) and FMDA-OT (Ghannou & Bennani, 2024), which operate without access to target data during training.

Table 1: Stability comparison between vanilla FedSCM vs. FedSCM+SizeReg (SR). Smaller values indicate more stable outputs.

| Method | D.R. | Hamming | Entropy | Cosine | F1 | Jaccard |
|---|---|---|---|---|---|---|
| FedSCM | 0.587 | 0.495 | 0.030 | 0.501 | 0.499 | 0.397 |
| FedSCM+SR | 0.540 | 0.484 | 0.015 | 0.480 | 0.479 | 0.381 |

Table 2: Performance comparison of various methods on CIFAR-10 and CIFAR-100 datasets for different values of $\alpha$.

| Dataset | CIFAR-10 | | | CIFAR-100 | | |
|---|---|---|---|---|---|---|
| $\alpha$ | 1.0 | 0.1 | 0.01 | 1.0 | 0.1 | 0.01 |
| FedAvg | $34.12 \pm 0.12$ | $29.40 \pm 0.10$ | $26.58 \pm 0.15$ | $15.26 \pm 0.21$ | $13.42 \pm 0.09$ | $4.34 \pm 0.01$ |
| K3DA | $64.27 \pm 0.02$ | $63.68 \pm 0.01$ | $44.74 \pm 0.02$ | $35.65 \pm 0.26$ | $37.45 \pm 0.18$ | $21.30 \pm 0.07$ |
| Co-MDA | $63.23 \pm 0.02$ | $60.17 \pm 0.02$ | $45.86 \pm 0.03$ | $33.32 \pm 0.01$ | $20.26 \pm 0.01$ | $21.01 \pm 0.00$ |
| FMDA-OT | $56.02 \pm 0.11$ | $20.43 \pm 0.03$ | $10.26 \pm 0.09$ | $18.67 \pm 0.08$ | $15.09 \pm 0.02$ | $5.68 \pm 0.01$ |
| SDEA | $23.01 \pm 0.10$ | $46.77 \pm 0.21$ | $23.9 \pm 0.14$ | $35.15 \pm 0.08$ | $9.04 \pm 0.5$ | $3.49 \pm 0.43$ |
| Source Free | $72.34 \pm 0.09$ | $47.62 \pm 0.22$ | $31.50 \pm 0.18$ | $26.08 \pm 0.23$ | $19.05 \pm 0.15$ | $9.73 \pm 0.24$ |
| Best Source | $69.67 \pm 0.00$ | $68.50 \pm 0.00$ | $53.44 \pm 0.00$ | $49.06 \pm 0.00$ | $43.14 \pm 0.00$ | $32.03 \pm 0.00$ |
| Random ($K = 3$) | $52.04 \pm 2.32$ | $25.66 \pm 7.41$ | $10.59 \pm 0.55$ | $8.08 \pm 3.94$ | $7.99 \pm 0.37$ | $4.47 \pm 2.59$ |
| Random ($K = 5$) | $57.77 \pm 3.35$ | $26.48 \pm 4.82$ | $15.63 \pm 6.25$ | $10.72 \pm 2.47$ | $7.25 \pm 1.55$ | $2.44 \pm 2.90$ |
| Random ($K = 10$) | $71.31 \pm 1.00$ | $62.17 \pm 2.10$ | $44.51 \pm 10.30$ | $30.61 \pm 0.60$ | $27.66 \pm 1.50$ | $11.61 \pm 0.90$ |
| **FedSCM** ($\gamma = 0.67$) | $77.43 \pm 0.11$ | $71.65 \pm 0.08$ | $\mathbf{54.61} \pm 0.14$ | $\mathbf{50.09} \pm 0.25$ | $\mathbf{47.71} \pm 0.18$ | $\mathbf{37.36} \pm 0.22$ |
| **FedSCM** ($\gamma = 0.8$) | $\mathbf{79.15} \pm 0.10$ | $\mathbf{75.51} \pm 0.15$ | $53.18 \pm 0.09$ | $41.33 \pm 0.31$ | $40.80 \pm 0.21$ | $36.41 \pm 0.17$ |
| **FedSCM** ($\gamma = 0.9$) | $75.87 \pm 0.20$ | $68.61 \pm 0.09$ | $33.18 \pm 0.32$ | $35.34 \pm 0.41$ | $32.30 \pm 0.26$ | $16.02 \pm 0.24$ |

- Federated Domain Adaptation (e.g., Co-MDA (Liu et al., 2023)), which performs domain adaptation via federated learning while still leveraging source data, and SDEA (Huang et al., 2024), which improves adaptation by selecting relevant clients based on minimum output entropy to emphasize confident target predictions.

- FedAvg (McMahan et al., 2017), a canonical FL algorithm that aggregates client updates without domain-specific adaptation.

- Random Selection, which selects clients uniformly at random to match the number of selected clients in our approach, serving as a fair baseline.

- Source-Free (One-Shot), which performs a single round of client selection without iterative federated updates.

- Our Method (FedSCM), evaluated with two different sparsity parameters: $\gamma = 0.67$ (selecting 3 clients), $\gamma = 0.8$ (selecting 5 clients), and $\gamma = 0.9$ (selecting 10 clients).

Tables 2 and 3 report the performance of our method on CIFAR-10, CIFAR-100, and SVHN under label and quantity shift, using $\alpha$ values of 1.0, 0.1, and 0.01 across 500 clients. Table 4 presents analogous results under covariate shift. These comparisons encompass a diverse set of existing approaches to contextualize the effectiveness of our client selection strategy. In all of our experiments, FedSCM outperforms existing methods by a significant margin which suggests its applicability over a wide range of federate learning tasks and various datasets.

Importantly, none of the existing methods are explicitly designed to handle the unique challenges posed by our setting: a large client pool where only a sparse subset possesses data distributions closely aligned with the target. Consequently, we have adapted representative techniques from related domains to serve as baselines within our evaluation framework. All experiments were repeated three times to ensure robustness, and the tables report averaged results. The primary evaluation metric is accuracy on the target dataset. To ensure a fair comparison in terms of computational budget, all methods were constrained to a maximum of 1000 optimization iterations.

Finally, Table 5 provides a summary of key characteristics of various existing methods, comparing them across several dimensions such as the presence of theoretical guarantees, the ability to handle unlabeled server-side datasets, and whether they fall under domain adaptation or domain generalization paradigms.

### 6.3.4 Hyperparameter Selection

The hyperparameter values were chosen via cross-validation. Our results are not highly sensitive to these choices, demonstrating the robustness of the proposed method. In Tables 2, 3, and 4, we report the performance of Our Method (FedSCM) with three different sparsity parameters: $\gamma = 0.67$ (selecting 3 clients), $\gamma = 0.8$ (selecting 5 clients) and $\gamma = 0.9$ (selecting 10 clients). These values of $\gamma$ were selected to balance sparsity and model performance.

**On the choice of $\gamma$ in practice without cross-validation.** The hyperparameter $\gamma \in (0, 1)$ controls the maximum number of selected clients via the sparsity bound $\lceil 1/(1 - \gamma) \rceil$ of Theorem 3. In our experiments, we set $\gamma$ based on the desired number of clients $s$ to be involved in the post-selection federated fine-tuning stage, via $\gamma = 1 - 1/s$, requiring no labeled target data. When no strong prior on $s$ is available, a practical alternative is to evaluate the FedSCM objective of Definition 2 — which depends only on the unlabeled server data $\mathcal{D}$ — across a small grid of $\gamma$ values and select the point at which the objective ceases to improve significantly, analogously to an elbow criterion.

### 6.3.5 Expertise Evaluation

To evaluate the validity of our modeling assumption regarding client model expertise, as introduced in Sections 4.2 and 4.3, we empirically estimated the expertise level of each client using the CIFAR-10 dataset

Table 3: Performance comparison of various methods on SVHN datasets for different values of $\alpha$.

| Dataset | SVHN | | |
|---|---|---|---|
| $\alpha$ | 1.0 | 0.1 | 0.01 |
| FedAvg | $22.59 \pm 0.20$ | $19.33 \pm 0.11$ | $20.39 \pm 0.05$ |
| K3DA | $60.44 \pm 0.08$ | $53.10 \pm 0.01$ | $31.30 \pm 0.01$ |
| Co-MDA | $60.03 \pm 0.07$ | $53.31 \pm 0.27$ | $32.53 \pm 0.15$ |
| FMDA-OT | $38.01 \pm 0.07$ | $29.69 \pm 0.12$ | $21.73 \pm 0.03$ |
| SDEA | $37.86 \pm 0.09$ | $19.58 \pm 0.26$ | $15.93 \pm 0.3$ |
| Source Free | $46.08 \pm 0.09$ | $38.75 \pm 0.16$ | $19.80 \pm 0.09$ |
| Best Source | $64.14 \pm 0.00$ | $58.38 \pm 0.00$ | $30.47 \pm 0.00$ |
| Random ($K = 3$) | $39.62 \pm 3.38$ | $15.31 \pm 6.46$ | $18.22 \pm 2.71$ |
| Random ($K = 5$) | $44.10 \pm 2.46$ | $20.05 \pm 4.81$ | $10.69 \pm 5.40$ |
| Random ($K = 10$) | $58.44 \pm 3.80$ | $43.12 \pm 2.66$ | $32.49 \pm 4.62$ |
| **FedSCM** ($\gamma = 0.67$) | $74.22 \pm 0.12$ | $67.11 \pm 0.12$ | $\mathbf{49.70} \pm 0.16$ |
| **FedSCM** ($\gamma = 0.8$) | $\mathbf{75.06} \pm 0.15$ | $\mathbf{67.81} \pm 0.09$ | $43.64 \pm 0.21$ |
| **FedSCM** ($\gamma = 0.9$) | $65.82 \pm 0.18$ | $52.43 \pm 0.07$ | $26.30 \pm 0.24$ |

Table 4: Comparative performance analysis of our model on CIFAR-10, CIFAR-100, and SVHN datasets with covariate shift, against various domain adaptation methods.

| Dataset | CIFAR-10 | CIFAR-100 | SVHN |
|---|---|---|---|
| FedAvg | $16.23 \pm 0.17$ | $5.70 \pm 0.21$ | $19.58 \pm 0.09$ |
| K3DA | $27.40 \pm 0.01$ | $6.29 \pm 0.01$ | $16.50 \pm 0.02$ |
| Co-MDA | $42.96 \pm 0.00$ | $15.14 \pm 0.00$ | $25.59 \pm 0.10$ |
| FMDA-OT | $15.48 \pm 0.06$ | $3.77 \pm 0.10$ | $19.55 \pm 0.02$ |
| SDEA | $10.48 \pm 0.19$ | $7.02 \pm 0.37$ | $19.58 \pm 0.11$ |
| Source Free | $56.18 \pm 0.25$ | $26.17 \pm 0.20$ | $16.98 \pm 0.12$ |
| Best Source | $39.13 \pm 0.00$ | $9.25 \pm 0.00$ | $29.53 \pm 0.00$ |
| **FedSCM** ($\gamma = 0.67$) | $\mathbf{60.31} \pm 0.24$ | $31.03 \pm 0.21$ | $\mathbf{28.41} \pm 0.30$ |
| **FedSCM** ($\gamma = 0.8$) | $59.85 \pm 0.34$ | $\mathbf{32.58} \pm 0.15$ | $25.36 \pm 0.22$ |

Table 5: Comparison of "key features" in several Federated Domain Adaptation (FDA) and/or Federated Domain Generalization (FDG) methods. The key features consist of User Scalability (US), Sparse Client Selection (SCS), Client Data Privacy Preservation (CDPP), Independency from Public Data (IPD), Iterative Refinement (IR), Theoretical Guarantees (TG), and Handling Unlabeled Target Datasets (HUTD).

| Method | US | SCS | CDPP | IPD | IR | TG | HUTD | Task |
|---|---|---|---|---|---|---|---|---|
| FedAvg (McMahan et al., 2017) | × | × | ✓ | ✓ | ✓ | × | ✓ | FDG |
| KD3A (Feng et al., 2021) | ✓ | × | × | × | × | × | ✓ | SFDA |
| FMDA-OT (Ghannou & Bennani, 2024) | × | × | ✓ | × | ✓ | × | ✓ | FDA |
| Co-MDA (Liu et al., 2023) | × | ✓ | ✓ | ✓ | ✓ | × | ✓ | FDA |
| SDEA (Huang et al., 2024) | ✓ | ✓ | ✓ | ✓ | ✓ | × | ✓ | FDA |
| **FedSCM (ours)** | ✓ | ✓ | ✓ | ✓ | ✓ | ✓ | ✓ | FDA |

from our earlier experiments. This estimation was performed by solving an optimization problem that fits the label distribution produced by a client's local model $h_k$ for any $k \in [K]$ on target samples in $\mathcal{D}$ to a two-component mixture: one component representing the true label distribution $P(\cdot|\boldsymbol{X})$ and the other representing Dirichlet noise. Specifically, the expertise level $\widehat{\eta}_k$ of client $k$ is defined as:

$$\widehat{\eta}_k \triangleq \underset{0 \leq \eta \leq 1}{\arg\min} \frac{1}{|\mathcal{D}|} \sum_{\boldsymbol{X} \in \mathcal{D}} \mathsf{KL}\left(h_k(\boldsymbol{X}) \middle\| \eta P(\cdot|\boldsymbol{X}) + (1-\eta)Q_{\boldsymbol{X}}\right),$$

where $h_k(\boldsymbol{X})$ denotes the predictive distribution of client $k$ on input $\boldsymbol{X}$, and $Q_{\boldsymbol{X}}$ is a Dirichlet-distributed random variable sampled independently for each input $\boldsymbol{X}$. The optimization was repeated multiple times with different random seeds regarding $Q$s, and results for $\widehat{\eta}_k$ were averaged once the variation across repetitions became negligible.

Figure 8 illustrates the relationship between the estimated expertise levels and the model accuracy on the target dataset. To provide additional insight, we also include a histogram of model accuracies across clients. The observed curve exhibits a strong correlation between expertise and accuracy, offering empirical support for the modeling framework proposed in Definition 4 and validating its applicability in practical scenarios. A possible explanation for this correlation is that when a local model $h_k$ (corresponding to client $k$) encounters relatively few samples from a particular class $c \in \{1, 2, \ldots, C\}$ during training, it tends to produce less reliable (and often random) predictions on unseen samples from that class. These predictions may vary in confidence, appearing as soft outputs with either high or low certainty, thus resembling a Dirichlet-distributed noisy measure.

# 7 Conclusions

In this work, we proposed **Federated Sparse Consensus Matching (FedSCM)**, an integrated optimization-driven federated learning framework designed to tackle the challenges of client selection and domain adaptation in heterogeneous, non-IID environments with sparsely relevant clients. FedSCM enables the server to efficiently identify and aggregate model updates only from a compact subset of clients whose data distributions align closely with the target domain, all while preserving client data privacy and without requiring server-side labels or detailed client statistics. We provided rigorous theoretical guarantees demonstrating the sparsity of client selection, accuracy of the resulting global model, and convergence of the optimization procedure. Extensive experiments on benchmark datasets subject to various distribution shifts showed that FedSCM consistently outperforms existing state-of-the-art federated learning and domain adaptation methods in both predictive performance and communication efficiency.

Our findings highlight the importance and effectiveness of principled, entropy-based client selection for scalable and robust federated learning in practical, privacy-sensitive applications. The FedSCM algorithm might demonstrate suboptimal performances in some scenarios such as: (i) when we have an insufficient number of clients with meaningfully-aligned data distributions, (ii) inadequate local dataset sizes relative to the model complexity, and (iii) fundamental mismatches between client and target meta-distributions. These

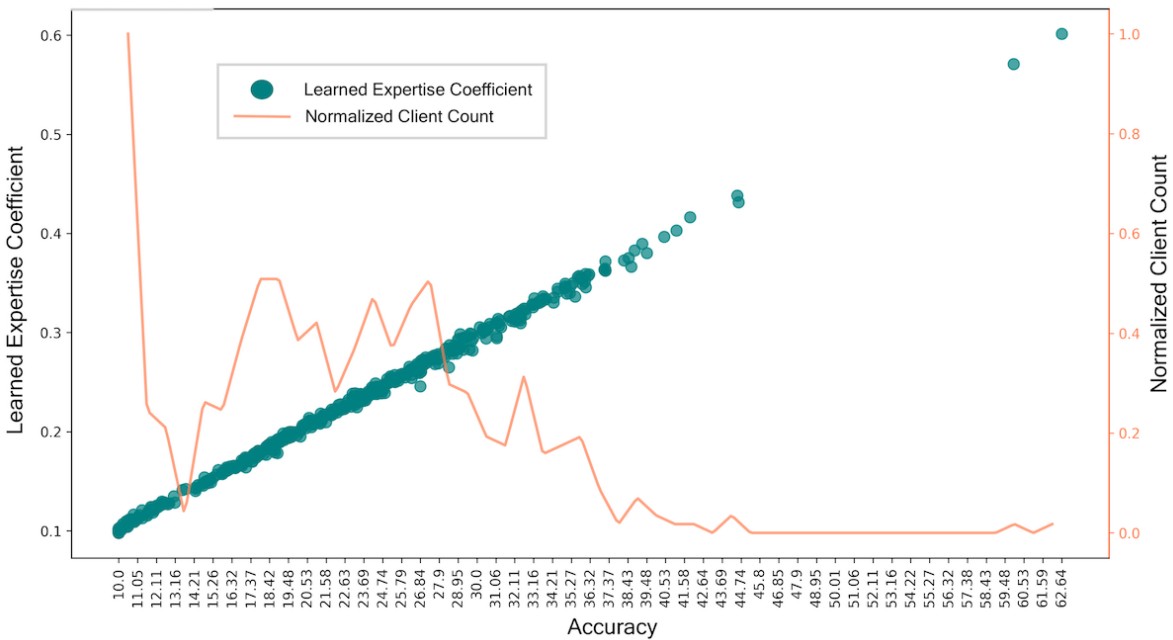

Figure 8: Relationship between a local model, denoted by $h_k$s, estimated expertise level $\widehat{\eta}_k$ (y-axis), and the model's accuracy (x-axis) on the target data. As can be seen, these two variables are strongly correlated.

limitations particularly impair knowledge transfer and model adaptation in heterogeneous environments, with overall system performance being highly sensitive to both proper $\gamma$ parameter selection and the availability of sufficiently many relevant clients. Future research directions include extending FedSCM to dynamic client populations and exploring adaptive optimization techniques to further enhance scalability and robustness under non-stationary data distributions.

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

## A    Proofs for Sparsity

*Proof of Theorem 3.* Recall that the objective of Definition 2 is

$$F(\boldsymbol{\lambda}) = \frac{1}{|\mathcal{D}|} \sum_{\boldsymbol{X} \in \mathcal{D}} \mathbb{H}\left(\sum_{k=1}^{K} \lambda_k h_k(\boldsymbol{X})\right),$$

where $\mathbb{H}$ denotes the Shannon entropy. A direct computation shows that the $(j, k)$-th entry of the Hessian of $F$ at any feasible $\boldsymbol{\lambda}$ is

$$H_{j,k}(\boldsymbol{\lambda}) = -\frac{1}{|\mathcal{D}|} \sum_{\boldsymbol{X} \in \mathcal{D}} \sum_{i=1}^{C} w_i(\boldsymbol{X}) \, h_j^{(i)}(\boldsymbol{X}) \, h_k^{(i)}(\boldsymbol{X}),$$

where $w_i(\boldsymbol{X}) = \left(\sum_{\ell=1}^{K} \lambda_\ell h_\ell^{(i)}(\boldsymbol{X})\right)^{-1} > 0$ for all $\boldsymbol{X}$ and all classes $i$ (see Proposition 8 for more details). For any $\boldsymbol{v} \in \mathbb{R}^K$, the associated quadratic form satisfies

$$\boldsymbol{v}^\top H(\boldsymbol{\lambda}) \, \boldsymbol{v} = -\frac{1}{|\mathcal{D}|} \sum_{\boldsymbol{X} \in \mathcal{D}} \sum_{i=1}^{C} w_i(\boldsymbol{X}) \left(\sum_{k=1}^{K} v_k \, h_k^{(i)}(\boldsymbol{X})\right)^2 \le 0,$$

so $H(\boldsymbol{\lambda}) \preceq 0$ and $F$ is concave everywhere on the feasible set. Now suppose $\boldsymbol{v} \ne \boldsymbol{0}$ lies in the tangent space of the simplex, i.e., $\sum_k v_k = 0$. The quadratic form above equals zero only if $\sum_k v_k h_k^{(i)}(\boldsymbol{X}) = 0$ for every $\boldsymbol{X} \in \mathcal{D}$ and every class $i \in [C]$ simultaneously. This is equivalent to $\boldsymbol{v}$ lying in the null space of the $nC \times K$ matrix formed by stacking the prediction vectors $h_k(\boldsymbol{X})$ over all $\boldsymbol{X} \in \mathcal{D}$ and all classes $i \in [C]$. By the full column rank assumption, this matrix has a trivial null space, so no such nonzero $\boldsymbol{v}$ exists. Therefore $H(\boldsymbol{\lambda}) \prec 0$ on the tangent space of the simplex, and $F$ is strictly concave on the feasible set.

Given the *strict* concavity of the objective, we start by a number of preliminary and simple results, and then prove the core result of the theorem. First, let us analyze the behavior of the entropy function $\mathbb{H}(\cdot)$ and leverage the properties of the convex sets established in the following lemmas.

**Lemma 12.** *The entropy function $\mathbb{H}(\boldsymbol{p})$ for a discrete probability measure $\boldsymbol{p} = (p_1, p_2, \ldots, p_n)$ is concave.*

*Proof.* To show that $\mathbb{H}(\boldsymbol{p})$ is concave, we need to consider the second derivative of $\mathbb{H}(\boldsymbol{p})$ with respect to each $p_i$ for $i \in [n]$. Let $f(p_i) = -p_i \log p_i$. The first and second derivatives of $f(p_i)$, respectively, are:

$$f'(p_i) = -\log p_i - 1 \qquad \text{and} \qquad f''(p_i) = -\frac{1}{p_i}. \tag{8}$$

Since $f''(p_i) \le 0$ for all $p_i > 0$, it indicates that $f(p_i)$ is concave. Because $\mathbb{H}(\boldsymbol{p})$ is the sum of concave functions $f(p_i)$, it follows that $\mathbb{H}(\boldsymbol{p})$ is concave. □

Due to the conservation of concavity via composition with linear/affine functions, the above lemma also proves that the summand $\mathbb{H}\left(\sum_{i=1}^{K} \lambda_i h_i(\boldsymbol{X})\right)$ in the objective of the program in Definition 2 is a concave function of $\lambda_i$s for $i \in [K]$.

**Lemma 13.** *Let $\Lambda = \{\boldsymbol{\lambda} \in \mathbb{R}^n \mid \sum_{i=1}^{n} \lambda_i = 1, \, 0 \le \lambda_i \le s \text{ for } i = 1, \ldots, n\}$, where $s > 0$. Then, the set $\Lambda$ is a convex set.*

*Proof.* By definition, a set $\Lambda$ is convex if, for any two points $\boldsymbol{\lambda}^{(1)}$ and $\boldsymbol{\lambda}^{(2)}$ in $\Lambda$ and any $\alpha \in [0, 1]$, the convex combination of $\boldsymbol{\lambda}^{(1)}$ and $\boldsymbol{\lambda}^{(2)}$ also belongs to $\Lambda$. Consider their convex combination:

$$\boldsymbol{\lambda} = \alpha \boldsymbol{\lambda}^{(1)} + (1 - \alpha) \boldsymbol{\lambda}^{(2)}, \quad \alpha \in [0, 1]. \tag{9}$$

We need to verify that $\boldsymbol{\lambda}$ satisfies the defining constraints of $\Lambda$.

Non-negativity of each of $\lambda_i$s for $i \in [n]$ is straightforward. Also, we can readily see that the sum condition holds as well, since:

$$\sum_{i=1}^{n} \lambda_i = \sum_{i=1}^{n} \left( \alpha \lambda_i^{(1)} + (1-\alpha)\lambda_i^{(2)} \right) = \alpha \sum_{i=1}^{n} \lambda_i^{(1)} + (1-\alpha) \sum_{i=1}^{n} \lambda_i^{(2)} = \alpha + (1-\alpha) = 1. \tag{10}$$

Second, we can show that the upper bound condition is satisfied via convex combination, i.e., for each $i = 1, \ldots, n$, we have:

$$\lambda_i = \alpha \lambda_i^{(1)} + (1-\alpha)\lambda_i^{(2)} \leq \alpha s + (1-\alpha)s = s. \tag{11}$$

Thus, $\lambda_i \leq s$ for all $i$. Since $\boldsymbol{\lambda}$ satisfies the conditions $\sum_{i=1}^{n} \lambda_i = 1$, $\lambda_i \geq 0$, and $\lambda_i \leq s$, it follows that $\boldsymbol{\lambda} \in \Lambda$ regardless of the choice of $\alpha$. Therefore, $\Lambda$ is a convex set. $\qquad\square$

From Lemma 12, we know that the entropy function $\mathbb{H}(\cdot)$ is concave. Also, Lemma 13 assures us that the feasible set of the constrained optimization problem in Definition 2, which forms the basis of our proposed approach is a convex set. This property implies that the minimum of the entropy function over a convex set occurs at the extreme points of that set. The following Lemma proves this argument.

**Lemma 14.** *Let $f : C \to \mathbb{R}$ be a strictly concave function defined on a convex set $C \subseteq \mathbb{R}^n$. Then, the minimum of $f$ on $C$ occurs at an extreme point of $C$.*

*Proof.* Let $C$ be a convex set and $f : C \to \mathbb{R}$ be a strictly concave function. Suppose $x^* \in C$ is a minimum of $f$ over $C$, i.e., $f(x^*) \leq f(x)$ for all $x \in C$. Assume, for the sake of contradiction, that $x^*$ is not an extreme point of $C$. Since $x^*$ is not an extreme point, it can be written as a convex combination of two distinct points $x$ and $y$ in $C$:

$$x^* = \alpha x + (1-\alpha)y, \quad \text{for some } \alpha \in (0,1) \text{ and } x, y \in C \text{ with } x \neq y. \tag{12}$$

By the definition of a concave function, for any $x, y \in C$ and $\alpha \in [0,1]$, we have:

$$f(\alpha x + (1-\alpha)y) \geq \alpha f(x) + (1-\alpha)f(y). \tag{13}$$

Substituting $x^* = \alpha x + (1-\alpha)y$ into this inequality, we get:

$$f(x^*) \geq \alpha f(x) + (1-\alpha)f(y). \tag{14}$$

Since $x^*$ is a global minimum of $f$ over $C$, we have $f(x^*) \leq f(x)$ and $f(x^*) \leq f(y)$. Letting $m = f(x^*)$, this implies $m \leq f(x)$ and $m \leq f(y)$. Thus, we have:

$$m \geq \alpha f(x) + (1-\alpha)f(y) \geq \alpha m + (1-\alpha)m = m. \tag{15}$$

The above inequality shows that:

$$f(x^*) = \alpha f(x) + (1-\alpha)f(y). \tag{16}$$

The equality condition in Jensen's inequality for strictly concave functions holds only if $f(x) = f(y) = f(x^*)$. Therefore, if $x^*$ is a minimum and is expressed as a convex combination of $x$ and $y$, then $f$ must take the same value at $x$, $y$, and $x^*$. This leads to a contradiction since a concave function, unless constant on the entire set, cannot have the same value at a point in the interior of $C$ and all points it is expressed as a combination of. If $f$ is not constant, then the minimum can only be achieved at the extreme points. Hence, the minimum concave function on a convex set occurs at an extreme point of the set $C$. $\qquad\square$

By Lemma 14, since $\mathbb{H}(\cdot)$ is concave, its value at the extreme points will give the minimum value over a convex set. At this point, we need to find the extreme points of the feasible set of the program in Definition 2, i.e., the convex set defined by the constraints

$$\Lambda = \left\{ \boldsymbol{\lambda} \in \mathbb{R}^n \,\middle|\, \sum_{i=1}^{n} \lambda_i = 1, \ 0 \leq \lambda_i \leq s \right\}. \tag{17}$$

$\Lambda$ has its extreme points characterized as follows: some $\lambda_i$ are equal to $s$, at most one $\lambda_i$ is between 0 and $s$, and the remaining $\lambda_i$'s are equal to 0. The following lemma proves this argument.

**Lemma 15.** *Let $\Lambda = \{\boldsymbol{\lambda} \in \mathbb{R}^n \mid \sum_{i=1}^n \lambda_i = 1, \ 0 \le \lambda_i \le s \ for \ i = 1, \dots, n\}$, where $s > 0$. The extreme points of $\Lambda$ occur when some $\lambda_i$'s are equal to $s$ (denote by $k$ the number of such $\lambda_i$'s), at most one $\lambda_i$ is between $0$ and $s$ (taking the remaining value $1 - ks$), and the other $\lambda_i$'s are equal to $0$. Also, we have*

$$k = \lceil 1/s \rceil.$$

*Proof.* Recall that an extreme point of a convex set is a point that cannot be expressed as a convex combination of any two distinct points in the set. Suppose, for the sake of contradiction, that there exist at least two $\lambda_i, \lambda_j$ in $\boldsymbol{\lambda} \in \Lambda$ such that $0 < \lambda_i < s$ and $0 < \lambda_j < s$. Consider two new points $\lambda^{(1)}$ and $\lambda^{(2)}$ defined as:

$$\lambda^{(1)} = (\lambda_1, \dots, \lambda_i + \epsilon, \dots, \lambda_j - \epsilon, \dots, \lambda_n),$$
$$\lambda^{(2)} = (\lambda_1, \dots, \lambda_i - \epsilon, \dots, \lambda_j + \epsilon, \dots, \lambda_n),$$

where $\epsilon > 0$ is a small positive number chosen such that $0 \le \lambda_i \pm \epsilon \le s$ and $0 \le \lambda_j \pm \epsilon \le s$. Now consider the convex combination of these two points:

$$\boldsymbol{\lambda} = \frac{1}{2}\boldsymbol{\lambda}^{(1)} + \frac{1}{2}\boldsymbol{\lambda}^{(2)}. \tag{18}$$

Expanding the convex combination, we get:

$$\boldsymbol{\lambda} = \left(\lambda_1, \dots, \frac{1}{2}(\lambda_i + \epsilon) + \frac{1}{2}(\lambda_i - \epsilon), \dots, \frac{1}{2}(\lambda_j - \epsilon) + \frac{1}{2}(\lambda_j + \epsilon), \dots, \lambda_n\right)$$
$$= (\lambda_1, \dots, \lambda_i, \dots, \lambda_j, \dots, \lambda_n). \tag{19}$$

Thus, $\boldsymbol{\lambda}$ can be written as a convex combination of $\boldsymbol{\lambda}^{(1)}$ and $\boldsymbol{\lambda}^{(2)}$, which are two distinct points in $\Lambda$. This shows that if at least two $\lambda_i$'s between 0 and $s$, then $\boldsymbol{\lambda}$ cannot be an extreme point of $\Lambda$. Therefore, at an extreme point, we can have at most one $\lambda_i$ between 0 and $s$. If such a $\lambda_i$ exists, it must take the value $1 - ks$, where $k$ is the number of $\lambda_i$'s equal to $s$. The other $\lambda_i$'s must be equal to 0 to satisfy the condition $\sum_{i=1}^n \lambda_i = 1$. This completes the proof. $\qquad\square$

Lemmas 12 to 15 lead us to the statement of the theorem, and thus complete the proof. $\qquad\square$

## B  Proofs for Client Selection Accuracy

*Proof of Theorem 7.* Let us give a sketch of proof before proceeding to formal statements where the technical intricacy might become cumbersome to comprehend. Roughly speaking, the proof consists of two parts:

- In the first part, we prove that choosing the non-zero coefficients from the set $\mathcal{I}^*$, i.e., the set of client indices with expertise levels greater than or equal to $1 - \delta$, with high probability (over the selection of $n$ samples in $\mathcal{D}$) results into a small objective:

$$\frac{1}{|\mathcal{D}|} \sum_{\boldsymbol{X} \in \mathcal{D}} \mathbb{H}\left(\sum_{i=1}^K \lambda_i h_i(\boldsymbol{X})\right) \le \left[1 + \log(C-1) + \log\left(\frac{4}{b+4\delta}\right)\right]\left(\frac{b}{4} + \delta\right) + \mathcal{O}\left(\max\{\delta, b\}^2\right)$$
$$\le 2(b+\delta)\log\left(\frac{1}{b+\delta}\right)\log C, \quad (\text{for } C \ge 2, \ \delta, b \le 1/2). \tag{20}$$

  In other words, objective is upper-bounded by $\tilde{\mathcal{O}}(b + \delta)$, where $\tilde{\mathcal{O}}(\cdot)$ hides poly-logarithmic dependencies.

- In the second phase, we prove that if the chosen set of indices $\mathcal{I}$ differs from $\mathcal{I}$, i.e., $|\mathcal{I}^* - \mathcal{I}| \geq 1$, then with probability at least $1 - \zeta$ (over the selection of clients and data samples in $\mathcal{D}$) one would get an excessively inflated objective. Mathematically, we would have (w.p. $\geq 1 - \zeta$)

$$\frac{1}{|\mathcal{D}|} \sum_{\boldsymbol{X} \in \mathcal{D}} \mathbb{H}\left(\sum_{i=1}^{K} \lambda_i h_i(\boldsymbol{X})\right) \geq \frac{1}{2} C \psi(C, \alpha) \left(\frac{1 - \delta'}{k^* + 1}\right)^2 = \mathcal{O}\left(\left[\frac{1 - \delta'}{k^*}\right]^2\right), \tag{21}$$

with $\psi(C, \alpha)$ being defined as

$$\psi(C, \alpha) \triangleq \frac{\Gamma(C\alpha)}{\Gamma(\alpha)\Gamma((C-1)\alpha)} \int_0^1 q^\alpha (1-q)^{C\alpha - \alpha - 1} \log \frac{1}{q} \mathrm{d}q.$$

It is straightforward to see how the combination of above results proves the theorem. In fact, given the parameters $\delta, \delta', b, k^*, C$ and $\alpha$ be chosen such that the following inequality holds:

$$2(b + \delta) \log\left(\frac{1}{b + \delta}\right) \log C < \frac{1}{2} C \psi(C, \alpha) \left(\frac{1 - \delta'}{k^* + 1}\right)^2, \tag{22}$$

which, for some properly defined constant $\Psi_{C, \alpha}$, is equivalent to

$$\frac{b + \delta}{(1 - \delta')^2} \log\left(\frac{1}{b + \delta}\right) < \frac{C \psi(C, \alpha)}{4 \log C (k^* + 1)^2} \leq \frac{\Psi_{C, \alpha}}{k^{*2}}, \tag{23}$$

ensures the existence of a threshold parameter $\gamma$ for the FedSCM of Definition 2 such that the support set of the minimizer, i.e., $\mathcal{I} \triangleq \{i | \lambda_i^* > 0\}$, satisfies $\mathcal{I} \subseteq \mathcal{I}^*$. Choosing the largest threshold parameter $\gamma$ (maximum number of selected clients) such that the objective function of FedSCM stays below $2 \log C(b + \delta) \log(1/(b + \delta))$ provably gives us exactly $\mathcal{I}^*$. However, it does require the knowledge of $\delta, \delta', b$ and $\alpha$ of the Dirichlet noise. We now turn to the formal proof of stages and two described above.

**Stage I:** Let us begin with the following lemma.

**Lemma 16.** *Let $\boldsymbol{p}$ be a probability measure over $\mathcal{Y}$, and assume for a sufficiently small $b \geq 0$, we have $\mathbb{H}(\boldsymbol{p}) \leq b$. Then, the following holds:*

$$\exists i \in [C] \;\Big|\; p_i \geq 1 - \frac{b}{4} - \mathcal{O}(b^2), \quad \text{and} \quad p_j \leq \frac{b}{4} + \mathcal{O}(b^2), \; \forall j \neq i. \tag{24}$$

*Conversely, if $\max_{i \in [C]} p_i \geq 1 - \beta$, for a sufficiently small $\beta \geq 0$, then*

$$\mathbb{H}(\boldsymbol{p}) \leq \beta \left(1 + \log(C - 1) + \log \frac{1}{\beta}\right) + \mathcal{O}(\beta^2). \tag{25}$$

*Proof.* Considering $\boldsymbol{p}$ as a probability measure over $\mathcal{Y} = [C]$, recall that the entropy of $\boldsymbol{p}$ can be written as $\mathbb{H}(\boldsymbol{p}) = -\sum_{i=1}^{C} p_i \log p_i$. Given that $b \geq 0$, $\mathbb{H}(\boldsymbol{p}) \leq b$, let us define $\beta$ such that $\max_i p_i = 1 - \beta$. The maximum occurs for at least one specific $i^* \in [C]$, thus we necessarily have $p_i \leq \beta$ for $i \neq i^*$ due to the fact that

$$p_{i^*} + \sum_{i \neq i^*} p_i = 1. \tag{26}$$

In the process of establishing a limiting bound for $\beta$ in terms of $b$, it is crucial to first determine the relation between the probability measure $\boldsymbol{p}$ and its entropy with $\beta$. The largest possible value for $\beta$, i.e., the minimum value for $\max_i p_i = 1 - \beta$, occurs when the entropy bound is touched and we have i) $\mathbb{H}(S) = b$, and ii) the smaller probabilities (corresponding to the remaining $C - 1$ classes) are distributed according the following rule: one dimension of $\boldsymbol{p}$ is exactly $\beta$, while the others (except for $i^*$) become zero. The reason behind this claim is as follows:

$$\mathbb{H}(\boldsymbol{p}) = (1 - \beta) \log \frac{1}{1 - \beta} + \sum_{i \neq i^*} p_i \log \frac{1}{p_i} \leq b. \tag{27}$$

Therefore, we have

$$(1 - \beta) \log \frac{1}{1 - \beta} \le b - \inf_\beta \; \sum_{i \ne i^*} p_i \log \frac{1}{p_i} = b - \beta \log \frac{1}{\beta}. \tag{28}$$

In this regard, it can be seen that this scenario is akin to a binary entropy setup for $\mathbb{H}(\boldsymbol{p})$ as below: $\mathbb{H}(\boldsymbol{p}) = -\beta \log(\beta) - (1 - \beta) \log(1 - \beta)$. Using a well-known inequality from Topsøe (2001), the above binary entropy has a lower bound of $4\beta(1 - \beta)$, which together with the assumption $\mathbb{H}(\boldsymbol{p}) \le b$ leads to

$$4\beta(1 - \beta) \le \mathbb{H}(S) \le b \;\Rightarrow\; 4\beta(1 - \beta) \le b \tag{29}$$

$$\Rightarrow\; \beta \le \frac{1 - \sqrt{1 - b}}{2} = \frac{b}{4} + \mathcal{O}\left(b^2\right).$$

It is evident that as $b$ approaches zero, indicating an increasingly concentrated distribution, $\beta$ must also approach zero.

Conversely, let us consider the scenario where $\beta \triangleq 1 - \max_{i \in [C]} p_i$ is sufficiently small. Our objective is to show that $\mathbb{H}(\boldsymbol{p})$ is also small. Note that the maximum entropy conditioned on $\beta$ is attained when the distribution of probabilities among the events is as uniform as possible: the remaining probability mass is distributed equally among the other $C - 1$ dimensions of $\boldsymbol{p}$. Thus, each will have a probability of $\frac{\beta}{C-1}$. The entropy of $S$ in this optimal dispersion case is given by:

$$\mathbb{H}(\boldsymbol{p}) \le (1 - \beta) \log \frac{1}{1 - \beta} + \sum_{i \ne i^*} \frac{\beta}{C - 1} \log \left(\frac{C - 1}{\beta}\right) \tag{30}$$

$$= (1 - \beta) \log \frac{1}{1 - \beta} + \beta \log \frac{C - 1}{\beta}$$

$$= (1 - \beta) \log \frac{1}{1 - \beta} + \beta \log (C - 1) + \beta \log \frac{1}{\beta}.$$

Moreover, we already know the following bounds exist for small enough $\beta \ge 0$:

$$(1 - \beta) \log \frac{1}{1 - \beta} \le \beta + \mathcal{O}\left(\beta^2\right). \tag{31}$$

Combining the above bounds will give us the claimed upper bound in the lemma and complete the proof. $\quad\square$

**Lemma 17.** *Assume $b \ge 0$ and $\delta \in (0, 1)$. Let $\boldsymbol{p}_1, \ldots, \boldsymbol{p}_k$ be $k \in \mathbb{N}$ probability measures over $[C]$, where for each $i \in [k]$ we have $\boldsymbol{p}_i = \eta_i P + (1 - \eta_i) Q_i$, with $P, Q_1, \ldots, Q_k \in \Delta^{C-1}$. Here, $P$ is assumed to be a fixed low-entropy measure with $\mathbb{H}(P) \le b$, and $Q_i$s are arbitrary distributions. Also, assume $\eta_i \ge 1 - \delta, \; \forall i \in [k]$. Then, the following bound holds for the average measure between $\boldsymbol{p}_i$s:*

$$\mathbb{H}\left(\sum_{i=1}^k \eta_i \boldsymbol{p}_i\right) \le \left[1 + \log(C - 1) + \log\left(\frac{4}{b + 4\delta}\right)\right] \left(\frac{b}{4} + \delta\right) + \mathcal{O}\left(\max\{\delta, b\}^2\right). \tag{32}$$

*Proof.* Using the result of Lemma 16, we have that the assumption $\mathbb{H}(P) \le b$ guarantees the presence of at least one label $i^*$ with probability $P(i^*) \ge 1 - \beta$, where $\beta \le b/4 + \mathcal{O}\left(b^2\right)$. Therefore, the following convex combination

$$\bar{\boldsymbol{p}} \triangleq \frac{1}{k} \sum_{i=1}^k \boldsymbol{p}_i = \left(\frac{1}{k} \sum_{i=1}^k \eta_i\right) P + \frac{1}{k} \sum_{i=1}^k (1 - \eta_i) Q_i$$

corresponds to a measure over $[C]$ where its maximum dimension exceeds

$$\max_i \bar{p}_i \ge 1 - \bar{\beta}, \quad \text{with} \quad \bar{\beta} \le 1 - (1 - \delta)\left(b/4 + \mathcal{O}(b^2)\right) = b/4 + \delta + \mathcal{O}\left(\max\{b, \delta\}^2\right).$$

Using the converse result from Lemma 16, we have

$$\mathbb{H}\left(\sum_{i=1}^{k}\eta_i\boldsymbol{p}_i\right) \le \left[1 + \log\left(C-1\right) + \log\left(\frac{4}{b+4\delta}\right)\right]\left(\frac{b}{4}+\delta\right) + \mathcal{O}\left(\max\{\delta,b\}^2\right) \tag{33}$$
$$= \tilde{\mathcal{O}}\left(b+\delta\right),$$

where $\tilde{\mathcal{O}}(\cdot)$ hides poly-logarithmic dependencies. This will complete the proof. $\qquad\square$

According to Assumption 6, for all $\boldsymbol{X} \in \mathcal{D} \sim P_{\mathcal{X}}^n$ we have $\mathbb{H}\left(P(\cdot|\boldsymbol{X})\right) \le b$. Therefore, the results of Lemmas 16 and 17 can be directly applied, and we have

$$\lambda_i^* = \left\{ \begin{array}{cc} 1/|\mathcal{I}^*| & i \in \mathcal{I}^* \\ 0 & \text{O.W.} \end{array} \right. \quad \Rightarrow$$
$$\frac{1}{|\mathcal{D}|}\sum_{\boldsymbol{X}\in\mathcal{D}}\mathbb{H}\left(\sum_{i=1}^{K}\lambda_i h_i\left(\boldsymbol{X}\right)\right) \le \left[1 + \log\left(C-1\right) + \log\left(\frac{4}{b+4\delta}\right)\right]\left(\frac{b}{4}+\delta\right) + \mathcal{O}\left(\max\{\delta,b\}^2\right)$$
$$= \tilde{\mathcal{O}}\left(b+\delta\right), \tag{34}$$

which indicates choosing $\mathcal{I}^*$ (or any of its subsets) will result into a *small* objective for the proposed SCM method. This concludes the stage I of the proof.

**Stage II:**  Again, we use the result of Theorem 3 and consider the $\lambda_i^*$ (coefficients corresponding to the minimizer of the FedSCM objective) to be $1/|\mathcal{I}|$ for $i \in \mathcal{I}$. In this regard, let us define $W$, i.e., the weighted average of expertise values of the client in $\mathcal{I}$, and subsequently bound it as follows:

$$W \triangleq \sum_{i\in\mathcal{I}}\lambda_i\eta_i = \sum_{i\in\mathcal{I}}\frac{\eta_i}{|\mathcal{I}|} = \sum_{i\in\mathcal{I}\cap\mathcal{I}^*}\frac{\eta_i}{|\mathcal{I}|} + \sum_{i\in\mathcal{I}-\mathcal{I}^*}\frac{\eta_i}{|\mathcal{I}|} \le 1 - \frac{|\mathcal{I}-\mathcal{I}^*|}{|\mathcal{I}|}(1-\delta'). \tag{35}$$

Also, note that we have $0 \le W \le 1$. We then use the fact that entropy is a concave function and hence, for any two measures $P, Q \in \Delta^{C-1}$ we have

$$\mathbb{H}\left(WP + (1-W)Q\right) \ge W\mathbb{H}\left(P\right) + (1-W)\mathbb{H}\left(Q\right). \tag{36}$$

Recall that for $\boldsymbol{X} \in \mathcal{D}$, we have $h_i(\boldsymbol{X}) = \eta_i P\left(\cdot|\boldsymbol{X}\right) + (1-\eta_i)Q_{\boldsymbol{X}}^{(i)}\left(\cdot\right)$ with $Q_{\boldsymbol{X}}^{(i)}\left(\cdot\right) \in \Delta^{C-1}$ representing an independent sample of a Dirichlet distribution Diriclet $(\boldsymbol{\alpha})$ with $\boldsymbol{\alpha} = (\alpha,\ldots,\alpha) \in \mathbb{R}^C$. Combining these two facts leads us to the following chain of inequalities:

$$\frac{1}{|\mathcal{D}|}\sum_{\boldsymbol{X}\in\mathcal{D}}\mathbb{H}\left(\sum_{i=1}^{K}\lambda_i h_i\left(\boldsymbol{X}\right)\right) \ge \frac{W}{|\mathcal{D}|}\sum_{\boldsymbol{X}\in\mathcal{D}}\mathbb{H}\left(\mathbb{P}\left(\cdot|\boldsymbol{X}\right)\right) + \frac{1-W}{|\mathcal{D}|}\sum_{\boldsymbol{X}\in\mathcal{D}}\mathbb{H}\left(\sum_{i\in\mathcal{I}}\left[\frac{\lambda_i(1-\eta_i)}{1-W}\right]Q_{\boldsymbol{X}}^{(i)}(\cdot)\right)$$
$$\ge \frac{|\mathcal{I}-\mathcal{I}^*|}{|\mathcal{I}|}(1-\delta')\cdot\frac{1}{|\mathcal{D}|}\sum_{\boldsymbol{X}\in\mathcal{D}}\mathbb{H}\left(\sum_{i\in\mathcal{I}}\theta_i Q_{\boldsymbol{X}}^{(i)}\right), \tag{37}$$

where we have used the fact that $\mathbb{H}\left(\mathbb{P}\left(\cdot|\boldsymbol{X}\right)\right) \ge 0$ for all $\boldsymbol{X} \in \mathcal{D}$. Also, we have defined $\theta_i$s as

$$\theta_i \triangleq \frac{\lambda_i(1-\eta_i)}{1-W}, \quad i \in \mathcal{I}.$$

Next, we use an important property of the finite Dirichlet distribution supported over $\Delta^{C-1}$: if $Q \sim$ Dirichlet $(\boldsymbol{\alpha})$, then each coordinate of $Q \in \Delta^{C-1}$ has a Beta $(\alpha, (C-1)\alpha)$ marginal distribution. Hence,

the following relation holds:

$$\mathbb{E}_{Q \sim \mathrm{Dirichlet}(\boldsymbol{\alpha})}\left(\mathbb{H}\left(Q\right)\right) = \mathbb{E}\left(\sum_{i=1}^{C} Q_i \log \frac{1}{Q_i}\right) \tag{38}$$

$$= \sum_{i=1}^{C} \mathbb{E}\left(Q_i \log \frac{1}{Q_i}\right)$$

$$= C \mathbb{E}_{q \sim \mathrm{Beta}(\alpha, (C-1)\alpha)}\left(q \log \frac{1}{q}\right).$$

So far, the lower-bound for the objective function of FedSCM has been reduced to the following formulation:

$$\frac{1}{|\mathcal{D}|} \sum_{\boldsymbol{X} \in \mathcal{D}} \mathbb{H}\left(\sum_{i=1}^{K} \lambda_i h_i\left(\boldsymbol{X}\right)\right) \geq \frac{C|\mathcal{I} - \mathcal{I}^*|}{|\mathcal{I}|}(1 - \delta')\left\{\frac{1}{|\mathcal{D}|} \sum_{\boldsymbol{X} \in \mathcal{D}} \bar{p}_{\boldsymbol{X}} \log \frac{1}{\bar{p}_{\boldsymbol{X}}}\right\}, \tag{39}$$

where $\bar{p}_{\boldsymbol{X}} \in [0,1]$ for $\boldsymbol{X} \in \mathcal{D}$ is defined as the first (or any other) component of the (weighted) aggregate of independent Dirichlet distributions, i.e.,

$$\bar{p}_{\boldsymbol{X}} \triangleq \sum_{i \in \mathcal{I}} \theta_i Q_{1,\boldsymbol{X}}^{(i)} = \sum_{i \in \mathcal{I}} \frac{\lambda_i(1 - \eta_i)}{1 - W} Q_{1,\boldsymbol{X}}^{(i)}. \tag{40}$$

It should be noted that the function $q \to q \log(1/q)$ is strictly concave over $q \in (0,1)$. Therefore, the following bound holds:

$$\bar{p}_{\boldsymbol{X}} \log \frac{1}{\bar{p}_{\boldsymbol{X}}} \geq \sum_{i \in \mathcal{I}} \frac{\lambda_i(1 - \eta_i)}{1 - W}\left[Q_{1,\boldsymbol{X}}^{(i)} \log \frac{1}{Q_{1,\boldsymbol{X}}^{(i)}}\right] \geq \sum_{i \in \mathcal{I} - \mathcal{I}^*} \frac{1 - \delta'}{|\mathcal{I}|}\left[Q_{1,\boldsymbol{X}}^{(i)} \log \frac{1}{Q_{1,\boldsymbol{X}}^{(i)}}\right]. \tag{41}$$

Recalling the fact that $Q_{1,\boldsymbol{X}}^{(i)}$ for each $i$ and $\boldsymbol{X} \in \mathcal{D}$ is an independent copy of a random variable distributed according to $\mathrm{Beta}(\alpha, (C-1)\alpha)$, and by defining $m \triangleq |\mathcal{D}| \cdot |\mathcal{I} - \mathcal{I}^*|$, we can conclude that

$$\mathbb{P}\left(\frac{1}{|\mathcal{D}|} \sum_{\boldsymbol{X} \in \mathcal{D}} \mathbb{H}\left(\sum_{i=1}^{K} \lambda_i h_i\left(\boldsymbol{X}\right)\right) \geq t\right) \tag{42}$$

$$\geq \mathbb{P}_{q_{1:m} \overset{i.i.d.}{\sim} \mathrm{Beta}(\alpha, (C-1)\alpha)}\left(C\left[\frac{|\mathcal{I} - \mathcal{I}^*|}{|\mathcal{I}|}(1 - \delta')\right]^2\left\{\frac{1}{m} \sum_{i=1}^{m} q_i \log \frac{1}{q_i}\right\} \geq t\right), \quad \forall t \in \mathbb{R}.$$

Therefore, we have managed to link the concentration properties of the objective function to that of $\frac{1}{m} \sum_{i=1}^{m} q_i \log \frac{1}{q_i}$ where $q_i$s are independent Beta-distributed random variables. In the next stage, we use the fact that each $q_i \log(1/q_i)$ is bounded and we can use Hoeffding-type inequalities of Sub-Gaussian random variables to bound the large deviation of the sum. The following simple lemma formalizes this approach:

**Lemma 18.** *Let $q \in [0,1]$ be a real-valued random variable, where $\mathbb{E}\left(-q \log q\right) = L > 0$. Then, for any $m \in \mathbb{N}$ and assuming $q_1, \ldots, q_m$ be $m$ independent copies of $q$, we have*

$$\mathbb{P}\left(\frac{1}{m} \sum_{i=1}^{m} q_i \log \frac{1}{q_i} \geq \frac{L}{2}\right) \geq 1 - e^{-mL^2/8}. \tag{43}$$

*Proof.* We have $0 \leq -q \log q \leq 2$ for all $q \in (0,1)$. Therefore, one can use the one-sided Hoeffding's inequality as follows:

$$\mathbb{P}\left(\frac{1}{m} \sum_{i=1}^{m} q_i \log \frac{1}{q_i} - \mathbb{E}\left[-q \log q\right] \leq -t\right) \leq e^{-mt^2/2}, \quad t \geq 0. \tag{44}$$

Setting $t = L/2$ proves the claim. $\qquad\square$

On the other hand, the following relation holds for the expected value $\mathbb{E}[q \log(1/q)]$:

$$\mathbb{E}\left[q \log \frac{1}{q}\right] = \int_0^1 q \log \frac{1}{q} \cdot \frac{\Gamma(C\alpha)}{\Gamma(\alpha)\Gamma((C-1)\alpha)} q^{\alpha-1}(1-q)^{(C-1)\alpha-1} \mathrm{d}q \tag{45}$$

$$= \frac{\Gamma(C\alpha)}{\Gamma(\alpha)\Gamma((C-1)\alpha)} \int_0^1 q^\alpha (1-q)^{C\alpha-\alpha-1} \log \frac{1}{q} \mathrm{d}q$$

$$\triangleq \psi(C, \alpha). \tag{46}$$

Using Lemma 18, the following high probability bound holds for the non-asymptotic lower-bound of the objective function for *a fixed* (non-random) selected subset $\mathcal{I} \subseteq [K]$:

$$\mathbb{P}\left(\frac{1}{|\mathcal{D}|} \sum_{\boldsymbol{X} \in \mathcal{D}} \mathbb{H}\left(\sum_{i=1}^K \lambda_i h_i(\boldsymbol{X})\right) \geq \frac{C\psi(C,\alpha)}{2}\left[\frac{|\mathcal{I} - \mathcal{I}^*|}{|\mathcal{I}|}(1-\delta')\right]^2\right) \geq 1 - e^{-n|\mathcal{I}-\mathcal{I}^*|\psi^2(C,\alpha)/8},$$

where we have defined $n \triangleq |\mathcal{D}|$ for the sake of simplicity in notations and clarity. Again, note that the above bound holds for a fixed $\mathcal{I} \subseteq [K]$, while the actual support set of $\boldsymbol{\lambda}^*$ (i.e., the minimizer of FedSCM objective) is a random set. Therefore, we need to extend the above bound to hold *uniformly* for all *bad* subsets of $[K]$, which are those subsets $\mathcal{I}$ with $|\mathcal{I} - \mathcal{I}^*| \geq 1$.

Before proceeding with the above agenda, it should be noted that we have

$$\frac{|\mathcal{I} - \mathcal{I}^*|}{|\mathcal{I}|} \geq \frac{1}{|\mathcal{I}^*| + 1} = \frac{1}{k^* + 1},$$

since $|\mathcal{I} - \mathcal{I}^*| \geq 1$ when there exists at least one wrongly selected client, and the minimum value for the nominator occurs when $|\mathcal{I}| = k^* + 1$, i.e., $\mathcal{I}$ is formed by the true client set $\mathcal{I}^*$ plus one addition irrelevant client.

The final step of the proof is devoted to enhancing the previous high probability bound to become uniform over all possible selected subsets $\mathcal{I}$ (with $|\mathcal{I} - \mathcal{I}^*| \geq 1$, i.e., all the sets that contain at least one unrelated client). First, the number of such sets $N$ can be attained as follows:

$$N = \sum_{|\Delta|=1}^{K-|\mathcal{I}^*|} \binom{K - |\mathcal{I}^*|}{|\Delta|}, \tag{47}$$

where $\Delta \triangleq \mathcal{I} - \mathcal{I}^*$. Based on *union bound*, we can write

$$\mathbb{P}\left(\forall \Delta, \ |\Delta| \geq 1 \ \middle| \ \frac{1}{n} \sum_{\boldsymbol{X} \in \mathcal{D}} \mathbb{H}\left(\sum_{i=1}^K \lambda_i h_i(\boldsymbol{X})\right) \geq \frac{1}{2} C\psi(C,\alpha)\left(\frac{1-\delta'}{k^*+1}\right)^2\right) \tag{48}$$

$$\geq 1 - \sum_{|\Delta|=1}^{K-|\mathcal{I}^*|} \binom{K - |\mathcal{I}^*|}{|\Delta|} e^{-n|\Delta|\psi^2(C,\alpha)/8}$$

$$\geq 1 - \sum_{i=1}^K \binom{K}{i} e^{-ni\psi^2(C,\alpha)/8}$$

$$\geq 1 - \sum_{i=1}^K \left(\frac{Ke}{i}\right)^i e^{-ni\psi^2(C,\alpha)/8} \qquad \text{since} \ \binom{K}{i} \leq \left(\frac{Ke}{i}\right)^i,$$

$$\geq 1 - \sum_{i=1}^K \left(\frac{Ke^{1-n\psi^2(C,\alpha)/8}}{i}\right)^i$$

$$\geq 1 - \sum_{i=1}^\infty \left(Ke^{1-n\psi^2(C,\alpha)/8}\right)^i$$

$$\geq 1 - \frac{Ke^{1-n\psi^2(C,\alpha)/8}}{1 - Ke^{1-n\psi^2(C,\alpha)/8}}. \tag{49}$$

Choosing $n$ large enough such that for $\zeta \in (0,1)$ we have

$$Ke^{1-n\psi^2(C,\alpha)/8} \leq \zeta/2 \quad \Rightarrow \quad n \geq n_{\min} \triangleq \frac{8\log\left(\frac{2Ke}{\zeta}\right)}{\psi^2(C,\alpha)},$$

results into the above lower-bound for the objective function to hold simultaneously for all possible subsets $\mathcal{I} \subseteq [K]$ with $|\mathcal{I} - \mathcal{I}^*| \geq 1$, with probability at least $1 - \delta$. In other words, for any $\zeta \in (0,1)$ and having $n \geq n_{\min}(K, C, \alpha, \zeta)$ unlabeled samples in the dataset $\mathcal{D}$, the FedSCM objective with probability at least $1 - \zeta$ takes a value of at least

$$\geq \frac{1}{2}C\psi(C,\alpha)\left(\frac{1-\delta'}{k^*+1}\right)^2$$

for any subset of clients $\mathcal{I}$ that have at least member which is not in $\mathcal{I}^*$. This completes the proof. $\qquad\square$

## B.1 Relaxing Assumption 6

Assumption 6 requires the Bayes conditional entropy $\mathbb{H}(P(\cdot \mid \boldsymbol{X}))$ to be bounded above by $b\log C$ for $P_{\mathcal{X}}$-almost every input $\boldsymbol{X}$. While this captures the intuition that the target problem is well-separated (i.e., the true label is nearly deterministic given the input for most samples), the almost-sure requirement can be overly restrictive in practice. Indeed, for any non-degenerate classification problem — such as a logistic model with a non-trivial decision boundary — there exist inputs $\boldsymbol{X}$ near the boundary for which $\mathbb{H}(P(\cdot \mid \boldsymbol{X}))$ approaches $\log C$, the maximum possible entropy. Requiring the bound $b < \log C$ to hold almost surely would therefore exclude even the simplest and most standard learning problems. It is thus natural to relax Assumption 6 to allow a small fraction of inputs to violate the entropy bound.

We consider the following two formulations:

(i) **Empirical fraction:** At least a $(1-\varrho)$-fraction of the $n$ samples in $\mathcal{D}$ satisfy $\mathbb{H}(P(\cdot \mid \boldsymbol{X})) \leq b$ with certainty, while the remaining $\varrho$-fraction may have entropy as large as $\log C$.

(ii) **Probabilistic:** The entropy bound holds with high probability over the input distribution, i.e.,

$$\mathbb{P}_{\boldsymbol{X} \sim P_{\mathcal{X}}}\left(\mathbb{H}(P(\cdot \mid \boldsymbol{X})) \leq b\right) \geq 1 - \varrho,$$

for some small $\varrho \in [0,1)$.

Formulation (i) is a statement about the finite sample $\mathcal{D}$, while formulation (ii) is a statement about the underlying distribution $P_{\mathcal{X}}$. By the law of large numbers, for large $n$ the empirical fraction of samples satisfying the bound concentrates around its population counterpart, so (i) and (ii) are essentially equivalent for large $n$. Formulation (i) is slightly easier to work with directly in the proof, and we adopt it below.

The proof of Theorem 7 carries through with only one modification. Under the original Assumption 6, the key intermediate bound on the objective evaluated at the oracle solution $\boldsymbol{\lambda}^* = \mathbf{1}_{\mathcal{S}^*}/|\mathcal{S}^*|$ (i.e., the uniform distribution over the truly relevant clients) is

$$\frac{1}{|\mathcal{D}|}\sum_{\boldsymbol{X}\in\mathcal{D}}\mathbb{H}\left(\sum_{i=1}^K \lambda_i^* h_i\left(\boldsymbol{X}\right)\right) \leq \left[1 + \log\left(C-1\right) + \log\left(\frac{4}{b+4\delta}\right)\right]\left(\frac{b}{4}+\delta\right) + \mathcal{O}\left(\max\{\delta,b\}^2\right)$$

$$= \tilde{\mathcal{O}}\left(b+\delta\right). \tag{50}$$

Under relaxation (i), the $(1-\varrho)$-fraction of samples satisfying $\mathbb{H}(P(\cdot|\boldsymbol{X})) \leq b$ contribute the same bound as before, while the remaining $\varrho$-fraction contribute at most $\log C$ each. The bound therefore becomes

$$\frac{1}{|\mathcal{D}|}\sum_{\boldsymbol{X}\in\mathcal{D}}\mathbb{H}\left(\sum_{i=1}^K \lambda_i^* h_i\left(\boldsymbol{X}\right)\right) \leq (1-\varrho)\left[1 + \log\left(C-1\right) + \log\left(\frac{4}{b+4\delta}\right)\right]\left(\frac{b}{4}+\delta\right) + \varrho\log C + \mathcal{O}\left(\max\{\delta,b\}^2\right)$$

$$= \tilde{\mathcal{O}}\left(b+\delta\right). \tag{51}$$

The rest of the proof proceeds identically to that of Theorem 7. Propagating equation 51 through the remainder of the argument, the final sample-complexity condition of Theorem 7 is modified to

$$(1 - \varrho)\frac{b + \delta}{(1 - \delta')^2} \log\left(\frac{1}{b + \delta}\right) + \frac{\varrho \log C}{(1 - \delta')^2} \leq \frac{\Psi_{C,\alpha}}{k^{*2}}, \tag{52}$$

replacing the original two-parameter condition on $(b, \delta, \delta')$.

The relaxed condition equation 52 now involves a four-tuple $(\varrho, b, \delta, \delta')$, each of which must be jointly small for the recovery guarantee to hold. Concretely: $b < \log C$ controls how well-separated the target problem is on the $(1 - \varrho)$-fraction of "easy" inputs; $\varrho$ controls how large the fraction of "hard" near-boundary inputs is allowed to be; $\delta$ captures the approximation quality of the client models on the easy inputs; and $\delta'$ controls the separation between relevant and irrelevant clients. When $\varrho = 0$, condition equation 52 reduces exactly to the original condition of Theorem 7, as expected. As $\varrho$ increases, the $\varrho \log C/(1-\delta')^2$ term grows, tightening the requirement on the remaining parameters $(b, \delta, \delta')$ to compensate. In the limit $\varrho \to 1$, the condition degenerates (the left-hand side approaches $\log C/(1 - \delta')^2$, which may exceed $\Psi_{C,\alpha}/k^{*2}$), reflecting the fact that if almost all inputs are near the decision boundary, the problem is genuinely hard and no selection method can be expected to succeed. The relaxation therefore does not trivialize the assumption — it simply acknowledges that a small fraction of hard inputs is unavoidable in practice, and quantifies precisely how much slack this introduces into the recovery condition.

## C   Proofs for Convergence

*Proof of Theorem 9.* Sequential Least Squares Quadratic Programming (SLSQP) works by solving a possibly constrained quadratic program at each iteration, in the following form:

$$\boldsymbol{\lambda}_{t+1}^* = \arg\min_{\boldsymbol{\lambda} \in \mathbb{R}^K} \ \frac{1}{2}\boldsymbol{\lambda}^\top H^{(t)}\boldsymbol{\lambda} + \boldsymbol{g}^{(t)\top}\boldsymbol{\lambda} \tag{53}$$
$$\text{subject to} \quad \boldsymbol{0} \preceq \boldsymbol{\lambda} \preceq (1 - \gamma)\boldsymbol{1},$$

where the fixed vector $\boldsymbol{g}^{(t)}$ and the matrix $H^{(t)} \in \mathbb{R}^{K \times K}$ represent the local linear and quadratic approximation of the objective at iteration $t$. In this way, SLSQP guarantees a reduction of the original objective over consecutive iterations.

We now prove that the output of the algorithm at each iteration $t \in \mathbb{N}$ lies at an extreme point of the polygonal feasible set $\left\{\boldsymbol{\lambda} \in \mathbb{R}^K \mid \boldsymbol{0} \preceq \boldsymbol{\lambda} \preceq (1 - \gamma)\boldsymbol{1}\right\}$. As shown in the proof of Theorem 3, such extreme points are at most $\lceil 1/(1 - \gamma)\rceil$-sparse. Since the objective is decreased at every iteration, the algorithm ultimately converges to such an extreme point, which is (at least) a local minimum of the FedSCM objective.

We begin by noting that since $H^{(t)}$, computed via Proposition 8, is negative definite, the quadratic objective $\frac{1}{2}\boldsymbol{\lambda}^\top H^{(t)}\boldsymbol{\lambda} + \boldsymbol{g}^{(t)\top}\boldsymbol{\lambda}$ is unbounded below, i.e., $\inf_{\boldsymbol{\lambda}} \mathsf{Obj}^{(t)}(\boldsymbol{\lambda}) = -\infty$. Based on this, we proceed by contradiction. Suppose that at some iteration $t$, two or more coordinates $\lambda_i$ are not at their respective box constraints, i.e., they are not equal to 0 or $1 - \gamma$. We show that in this case, the QP objective becomes unbounded below. Without loss of generality, assume that dimensions 1 and 2 are such inactive coordinates. That is, $\lambda_1$ and $\lambda_2$ lie strictly within their bounds. The QP constraint requires:

$$\sum_{i=1}^K \lambda_i = 1 \quad \Rightarrow \quad \lambda_1 + \lambda_2 + \sum_{i=3}^K \lambda_i = 1.$$

From standard results in convex optimization and duality, inactive box constraints can be temporarily removed without changing the optimal value. Assume all other $\lambda_i$ for $i \geq 3$ are fixed and satisfy their box constraints, such that $\sum_{i=3}^K \lambda_i = c$ is a fixed constant. Then:

$$\lambda_1 + \lambda_2 = 1 - c = \text{constant}.$$

Define a direction:

$$\lambda_1 = \alpha, \quad \lambda_2 = (1 - c) - \alpha, \quad \lambda_i = \text{fixed for } i \geq 3.$$

Hence, $\boldsymbol{\lambda}(\alpha)$ is feasible and satisfies the linear constraint (with the box constraints for $\lambda_1$ and $\lambda_2$ removed). Define

$$\mathbf{v} = [1, -1, 0, \ldots, 0]^\top.$$

Then, $\boldsymbol{\lambda}(\alpha) = \alpha\mathbf{v} + \boldsymbol{\lambda}_{\text{fixed}}$, and the QP objective becomes:

$$\mathsf{Obj}^{(t)}(\boldsymbol{\lambda}(\alpha)) = \frac{1}{2}\alpha^2\mathbf{v}^\top H^{(t)}\mathbf{v} + \alpha\mathbf{v}^\top \boldsymbol{g}^{(t)} + \text{const.}$$

Since $H^{(t)} \prec 0$ under the assumption of the theorem (see the proof of Theorem 3 for more details), we have $\mathbf{v}^\top H^{(t)}\mathbf{v} < 0$, and thus the objective diverges to $-\infty$ as $|\alpha| \to \infty$. However, this is impossible given the bounded feasible region defined by the box constraints. Therefore, our assumption must be false: at most one coordinate can lie strictly inside its box constraints at optimality. Hence, the solution must lie at an extreme point, completing the proof. □

*Proof of Theorem 10.* The proof is based on the following key observation: a simple analysis of the gradient formulation $\boldsymbol{g}$ from Proposition 8 reveals that when $\lambda_k$ increases locally at some step $t$ for a fixed $k \in [K]$, the corresponding gradient component $g_k^{(t+1)}$ decreases (i.e., becomes more negative). This implies a sharper increase in $\lambda_k$ in the next iteration, and this pattern continues until convergence. Thus, optimization algorithms that rely purely on first-order information—such as PGD (but unlike second-order methods like SLSQP)—are highly sensitive to the choice of initialization.

We initialize $\lambda_k^{(0)} = 1/K$ for all $k \in [K]$ and show that, under the assumptions in the theorem, the initial update of PGD results in a sharper increase in $\lambda_k$ for $k \in \mathcal{I}^*$ compared to those for $k \notin \mathcal{I}^*$. This sharper growth persists throughout the iterations, leading to the final selection of the highest-expertise clients, thereby completing the proof.

The following parts provide a detailed analysis of this process. Under the Dirichlet noise model of Definition 4, and assuming client $k \in [K]$ has an expertise level $\eta_k$ with respect to the true (latent) conditional distribution $P(\cdot|\boldsymbol{X})$, we establish:

$$\mathsf{KL}\left(h_k(\boldsymbol{X})\,\bigg\|\,\sum_{i=1}^K \lambda_i h_i(\boldsymbol{X})\right) = \mathsf{KL}\left(\eta_k P(\cdot|\boldsymbol{X}) + (1-\eta_k)Q_{k,\boldsymbol{X}}\,\bigg\|\,\bar{\eta}P(\cdot|\boldsymbol{X}) + (1-\bar{\eta})\bar{Q}\right), \quad \text{and}$$

$$\mathbb{H}\left(h_k(\boldsymbol{X})\right) = \mathbb{H}\left(\eta_k P(\cdot|\boldsymbol{X}) + (1-\eta_k)Q_{k,\boldsymbol{X}}\right), \tag{54}$$

where we have assumed initial weights $\lambda_k^{(0)}$ are all set to $1/K$ for $k \in [K]$, and random $C$-dimensional distributions $\Delta$ and $\bar{Q}$ are defined as

$$Q_{k,\boldsymbol{X}} \sim \text{Dirichlet}\left(\alpha, \ldots, \alpha\right),$$

$$\bar{Q} = \bar{Q}(\boldsymbol{X}) \triangleq \frac{1}{K(1-\bar{\eta})}\sum_{k\in[K]}(1-\eta_k)Q_{k,\boldsymbol{X}}, \tag{55}$$

with $\bar{\eta} \triangleq \sum_{k\in[K]}\eta_k/K$.

**Lemma 19.** *For any $c > 0$, with probability at least $1 - e^{-c^2}$ with respect to the randomness of drawing Dirichlet noise measures $Q_{k,\boldsymbol{X}} \in \Delta^{C-1}$, $k \in [K]$, we have*

$$\bar{Q}_i(\boldsymbol{X}) = \frac{1}{C} \pm c\sqrt{\frac{2\log(C|\mathcal{D}|)}{K(1-\bar{\eta})}}, \quad \forall i \in [C], \ \forall \boldsymbol{X} \in \mathcal{D},$$

*where $\bar{Q}(\boldsymbol{X}) \in \Delta^{C-1}$ is defined in equation 55.*

*Proof.* For any $i \in [C]$, $[Q_{k,\boldsymbol{X}}]_i$ (for all $k, \boldsymbol{X}$) is marginally distributed according to $\text{Beta}(\alpha, (C-1)\alpha)$, where $\alpha$ is the Dirichlet noise parameter. Therefore, it corresponds to a random variable distributed in $[0, 1]$ with mean $1/C$.

For any $\boldsymbol{X}$, each $\bar{Q}_i(\boldsymbol{X})$ is the weighted average of $K$ i.i.d. Beta-distributed R.V.s, where the $k$th weight is $1 - \eta_k$. Using Hoeffding's inequality, we have

$$\mathbb{P}\left(\left|\frac{1}{K(1-\bar{\eta})}\sum_{k=1}^{K}(1-\eta_k)\left[Q_{k,\boldsymbol{X}}\right]_i - \frac{1}{C}\right| \leq \varepsilon\right) \geq 1 - \exp\left(\frac{-K^2(1-\bar{\eta})^2\varepsilon^2}{2\sum_{k\in[K]}(1-\eta_k)^2}\right). \tag{56}$$

Since we have $K(1-\bar{\eta}) = \sum_{k\in[K]}(1-\eta_k) \geq \sum_{k\in[K]}(1-\eta_k)^2$, the inequality can be reformulated as

$$\mathbb{P}\left(\left|\bar{Q}_i(\boldsymbol{X}) - \frac{1}{C}\right| \leq \varepsilon\right) \geq 1 - e^{-K(1-\bar{\eta})\varepsilon^2/2}, \quad \forall i \in [C],\ \forall \boldsymbol{X} \in \mathcal{D}. \tag{57}$$

Using union bound, we can turn the above *point-wise* probabilistic inequality into a *uniform* equivalent as follows:

$$\mathbb{P}\left(\left|\bar{Q}_i(\boldsymbol{X}) - \frac{1}{C}\right| \leq \varepsilon\ \Big|\ \forall i \in [C],\ \forall \boldsymbol{X} \in \mathcal{D}\right) \geq 1 - C|\mathcal{D}|e^{-K(1-\bar{\eta})\varepsilon^2/2}. \tag{58}$$

Choosing

$$\varepsilon \triangleq c\sqrt{\frac{2\log(C|\mathcal{D}|)}{K(1-\bar{\eta})}}$$

completes the proof. $\qquad\square$

Combining the above results and using Lemma 8, for the $k$th component of the gradient (i.e., $g_k(\boldsymbol{\lambda}^{(0)})$), we have

$$g_k = \widehat{\mathbb{E}}_{\boldsymbol{X}\in\mathcal{D}}\left[\sum_{i\in[C]}\left(\eta_k P(i|\boldsymbol{X}) + (1-\eta_k)Q_{k,\boldsymbol{X}}(i)\right)\log\frac{1}{\bar{\eta}P(i|\boldsymbol{X}) + (1-\bar{\eta})\bar{Q}_i(\boldsymbol{X})}\right] - 1. \tag{59}$$

According to Lemma 19 and the Mean Value Theorem (MVT), for any $\zeta \in (0,1)$ the following upper and lower-bound holds with probability at least $1 - \zeta/2$ for all $k \in [K]$ and $\boldsymbol{X} \in \mathcal{D}$:

$$\log\frac{1}{\bar{\eta}P(i|\boldsymbol{X}) + (1-\bar{\eta})\bar{Q}_i(\boldsymbol{X})} = \log\frac{C}{1-\bar{\eta} + C\bar{\eta}P(i|\boldsymbol{X})} \pm C\sqrt{\frac{2\log(C|\mathcal{D}|)}{K(1-\bar{\eta})}\log\frac{2}{\zeta}}.$$

On the other hand, each $Q_{k,\boldsymbol{X}}(i)$ for $k \in [K]$, $\boldsymbol{X} \in \mathcal{D}$, and $i \in [C]$ is a Beta$(\alpha, (C-1)\alpha)$-distributed random variable with mean $1/C$. Using Hoeffding's inequality and union bound, similar to Lemma 18, we have

$$\mathbb{P}\left(\left|\widehat{\mathbb{E}}_{\boldsymbol{X}\in\mathcal{D}}\left[Q_{k,\boldsymbol{X}}(i)\right] - \frac{1}{C}\right| \leq \varepsilon\right) \geq 1 - KCe^{-|\mathcal{D}|\varepsilon^2/2}, \tag{60}$$

for any $\varepsilon \geq 0$. Alternatively, we can say that for any $\zeta \in (0,1)$, the following holds uniformly with probability at leat $1 - \zeta/2$:

$$\widehat{\mathbb{E}}_{\boldsymbol{X}\in\mathcal{D}}\left[Q_{k,\boldsymbol{X}}(i)\right] = \frac{1}{C} \pm \sqrt{\frac{2\log(KC)}{|\mathcal{D}|}\log\frac{2}{\zeta}}, \quad \forall k, \boldsymbol{X}, i. \tag{61}$$

Combining all the results so far, we get the following result with probability at least $1 - \zeta$ for any $\zeta \in (0,1)$ and uniformly for all $k \in [K]$:

$$g_k = -\eta_k\widehat{\mathbb{E}}_{\boldsymbol{X}\in\mathcal{D}}\left[\sum_{i\in[C]}\left(\frac{1}{C} - P(i|\boldsymbol{X})\right)\log\frac{1}{1-\bar{\eta} + C\bar{\eta}P(i|\boldsymbol{X})}\right] + \text{const.} \tag{62}$$

$$\pm\left(C\sqrt{\frac{2\log(C|\mathcal{D}|)}{K(1-\bar{\eta})}\log\frac{2}{\zeta}} + (1-\eta_k)\sqrt{\frac{2\log(KC)}{|\mathcal{D}|}\log\frac{2}{\zeta}}\log\frac{C}{1-\bar{\eta}} + \mathcal{O}\left((|\mathcal{D}|K)^{-1/2}\right)\right),$$

where const. refers to a constant value independent of $k$, and $\mathcal{O}\left((|\mathcal{D}|K)^{-1/2}\right)$ becomes negligible for sufficiently large $K$ and $|\mathcal{D}|$. Also, it should be noted that we have

$$\mathbb{H}\left(P(\cdot|\boldsymbol{X})\right) \overset{a.s.}{\leq} b$$

with respect to the randomness of $\boldsymbol{X}$. Now, for $\bar{\eta} \in (0,1)$, $C \geq 2$ and $b < \log C$, let us define $\theta(b, C, \bar{\eta})$ as

$$\theta(b, C, \bar{\eta}) \triangleq \inf_{\boldsymbol{p} \in \Delta^{C-1}} \sum_{i=1}^{C} (1/C - p_i) \log \frac{1}{1 - \bar{\eta} + C\bar{\eta}p_i}$$

$$\text{subject to} \quad \sum_{i=1}^{C} p_i \log \frac{1}{p_i} \leq b. \tag{63}$$

Therefore, we have

$$\widehat{\mathbb{E}}_{\boldsymbol{X} \in \mathcal{D}}\left[\sum_{i \in [C]} \left(\frac{1}{C} - P(i|\boldsymbol{X})\right) \log \frac{1}{1 - \bar{\eta} + C\bar{\eta}P(i|\boldsymbol{X})}\right] \overset{a.s.}{\geq} \theta(b, C, \bar{\eta}) > 0. \tag{64}$$

It is straightforward to see that the following extreme equalities hold for function $\theta$: $\theta(b = \log C, C, \bar{\eta}) = 0$, $\theta(b, C, \bar{\eta} = 0) = 0$. Also, it can be readily seen that increases logarithmically as $1 - \bar{\eta}$ and/or $b$ are being decreased, and also increases logarithmically with $C$. For some $\varepsilon > 0$, assume the following bounds hold:

$$C\sqrt{\frac{2\log(C|\mathcal{D}|)}{K(1 - \bar{\eta})} \log \frac{2}{\zeta}} \leq \frac{\varepsilon}{2} \implies \frac{K}{\log(C|\mathcal{D}|)} \geq \frac{8C^2}{\varepsilon^2(1 - \bar{\eta})} \log \frac{2}{\zeta},$$

$$\sqrt{\frac{2\log(KC)}{|\mathcal{D}|} \log \frac{2}{\zeta}} \log \frac{C}{1 - \bar{\eta}} \leq \frac{\varepsilon}{2} \implies \frac{|\mathcal{D}|}{\log(KC)} \geq \frac{8\log^2\left(\frac{C}{1-\bar{\eta}}\right)}{\varepsilon^2} \log \frac{2}{\zeta}. \tag{65}$$

Therefore, with probability at least $1 - \zeta$, for any $k, k' \in [K]$ with $k \neq k'$, we have

$$\frac{g_k^{(0)} - g_{k'}^{(0)}}{\eta_k - \eta_{k'}} \leq -\theta(b, C, \bar{\eta}) + \frac{2\varepsilon}{|\eta_k - \eta_{k'}|}. \tag{66}$$

Assuming $k \in \mathcal{I}^*$ and $k' \notin \mathcal{I}^*$, the term $|\eta_k - \eta_{k'}|$ is lower-bounded by $1 - (\delta + \delta')$. Therefore, choosing

$$\varepsilon \triangleq \frac{\theta(b, C, \bar{\eta})}{2}\left(1 - (\delta + \delta')\right)$$

guarantees the following:

$$g_k^{(0)} - g_{k'}^{(0)} < 0, \quad \forall k \in \mathcal{I}^*, \ k' \notin \mathcal{I}^*. \tag{67}$$

Thus, during the gradient descent iterations of Algorithm 1, the client coefficients $\lambda_k$ for $k \in \mathcal{I}^*$ increase (with probability at least $1 - \zeta$) more than those for $k \notin \mathcal{I}^*$, particularly in the initial step when all $\lambda_k$ values are initialized at $1/K$. Based on the argument presented at the beginning of the proof, this guarantees that the coefficients corresponding to $k \in \mathcal{I}^*$ are the first to reach $1 - \gamma$ during the PGD iterations. Substituting the previously derived $\varepsilon$ into the inequalities for $K$ and $|\mathcal{D}|$ from equation 65, we obtain the refined bounds:

$$\frac{|\mathcal{D}|}{\log(KC)} \geq \frac{32\log^2\left(\frac{C}{1-\bar{\eta}}\right)}{\theta^2(b, C, \bar{\eta})} \frac{\log \frac{2}{\zeta}}{(1 - (\delta + \delta'))^2} \triangleq \frac{\theta_1(b, C, \bar{\eta})}{(1 - (\delta + \delta'))^2} \log \frac{2}{\zeta},$$

$$\frac{K}{\log(C|\mathcal{D}|)} \geq \frac{32C^2}{(1 - \bar{\eta})\theta^2(b, C, \bar{\eta})} \frac{\log \frac{2}{\zeta}}{(1 - (\delta + \delta'))^2} \triangleq \frac{\theta_2(b, C, \bar{\eta})}{(1 - (\delta + \delta'))^2} \log \frac{2}{\zeta}. \tag{68}$$

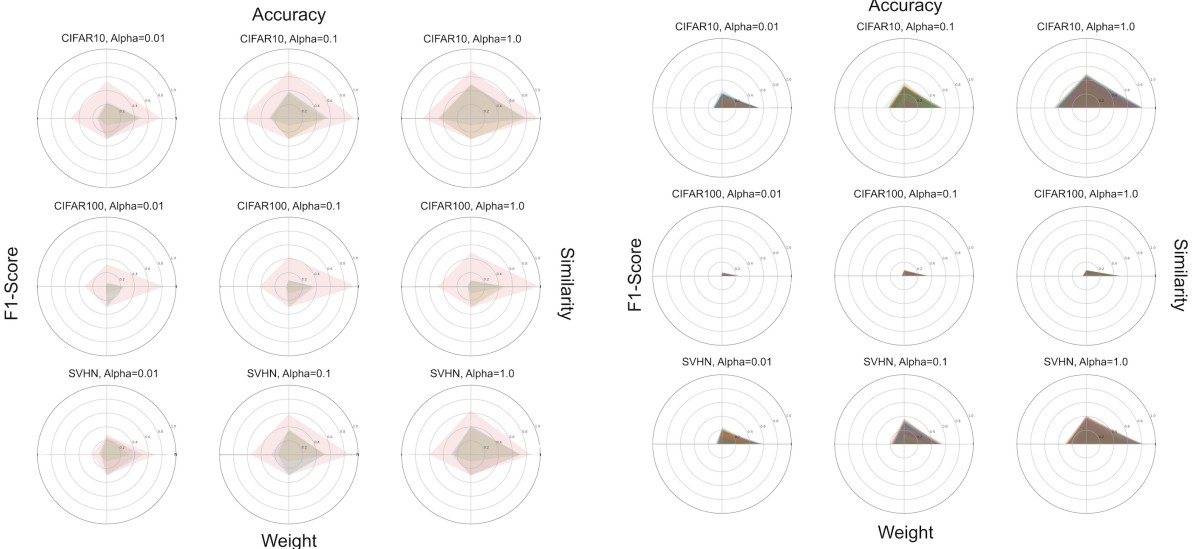

Figure 9: (Left) Radar plot showing the performance metrics (accuracy, F1 score, and data similarity) of selected clients (selecting 3 relevent clients). The plot highlights the superior performance characteristics of the selected clients. (Right) Radar plot for unselected clients, showing key metrics such as accuracy, F1 score, and data similarity. The unselected clients typically exhibit lower performance compared to the selected ones, emphasizing the effectiveness of the client selection process.

Additionally, note that the pathological case of $\bar{\eta} = 1$ is naturally handled by the original gradient formulation in Lemma 8. When $\bar{\eta} = 1$, there is no Dirichlet noise contamination, meaning all clients are optimal choices. In this scenario, the problem becomes ill-posed, as there is no meaningful selection process to be performed. This confirms the convergence to the global optimum of the objective in Definition 2.

To determine the rate of convergence, consider the gradient difference between an expert client $k \in \mathcal{I}^*$ and a non-expert client $k' \notin \mathcal{I}^*$. This difference, given by $g_k^{(t)} - g_{k'}^{(t)}$, is strictly negative with a magnitude of at least $-\theta(b, C, \bar{\eta})/2$ from $t = 0$, and it decreases as $t$ increases. Since the total variation in $\lambda_k$ is bounded by 1, the algorithm terminates in $\mathcal{O}(1/\theta)$ time, which remains constant when $b, C$, and $\bar{\eta}$ are fixed.

This completes the proof.

$\square$

# D  Auxiliary Experiments

The complete experimental results for client selection on CIFAR-10, CIFAR-100, and SVHN datasets—across Dirichlet heterogeneity parameters $\alpha \in \{0.01, 0.1, 1.0\}$—are provided in Figures 9, 10, 11 and 12. These figures aim to provide a comprehensive evaluation of the behavior and effectiveness of the proposed FedSCM method in identifying high-quality clients for source-free domain adaptation under varying degrees of non-IID data distributions.

Figures 9 and 10 (left) and (right) feature radar plots that compare the characteristics of selected versus unselected clients along several key performance dimensions. The axes of these plots include metrics such as classification accuracy on the server's unlabeled target dataset (estimated via pseudo-labeling), F1 score, and a proxy measure of data similarity to the target distribution, computed via prediction consistency. These visualizations help distinguish the performance profiles of the clients retained by FedSCM from those excluded by the selection process. Notably, selected clients tend to exhibit not only higher accuracy but also more stable and coherent predictions with respect to the target distribution, highlighting the method's ability to detect clients with useful domain alignment despite the absence of labeled data.

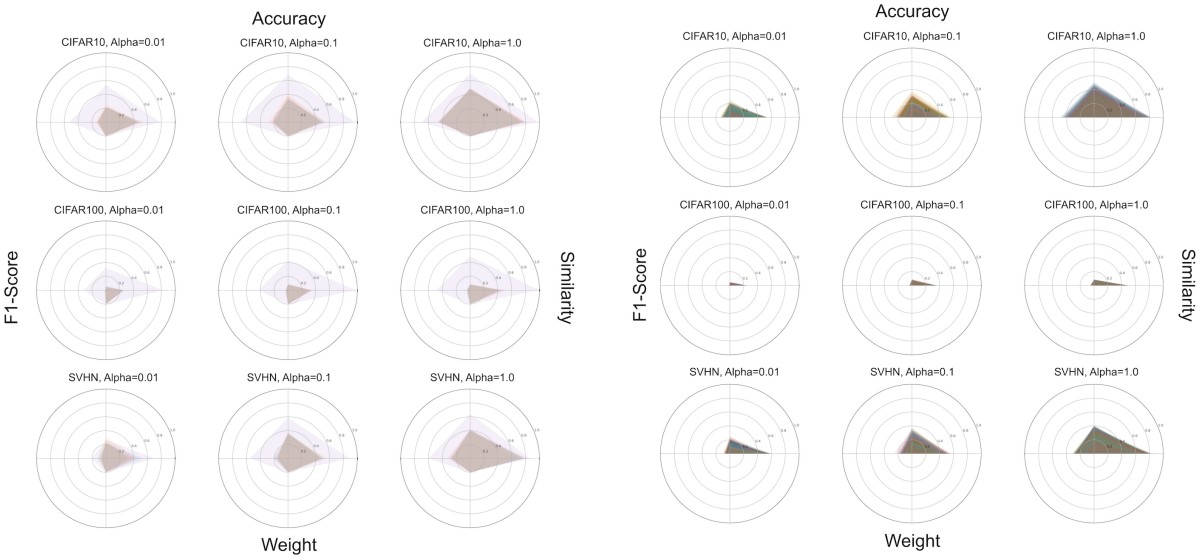

Figure 10: On the left, a radar plot illustrates the performance metrics—accuracy, F1 score, and data similarity—for a representative subset of five selected clients, showcasing their strong performance. In contrast, the right-hand radar plot displays the same metrics for the unselected clients, who generally demonstrate lower performance. This visual comparison underscores the efficacy of the client selection strategy.

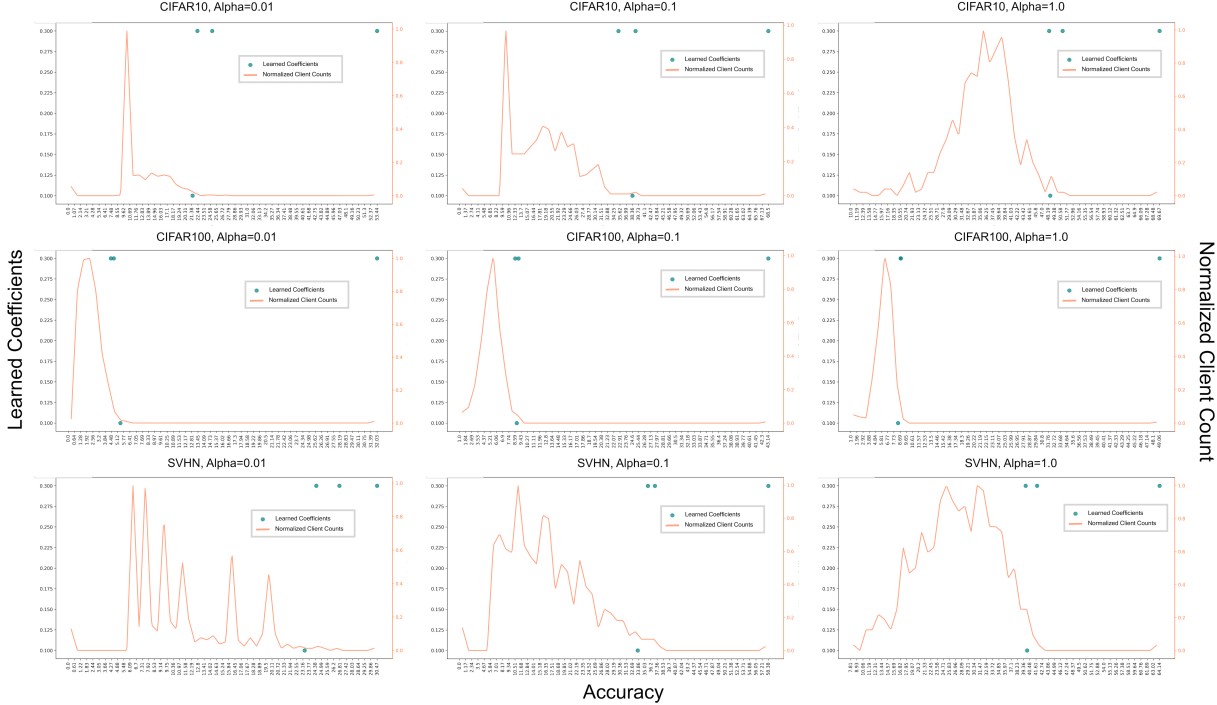

Figure 11: Sparsity analysis of client selection. The green plot shows the correlation between client accuracy and selection weights, while the orange plot presents the distribution of client accuracies (select 3 relevent clients). This figure underscores the sparsity of selected clients and demonstrates that the chosen clients are concentrated in the high-accuracy region of the dataset.

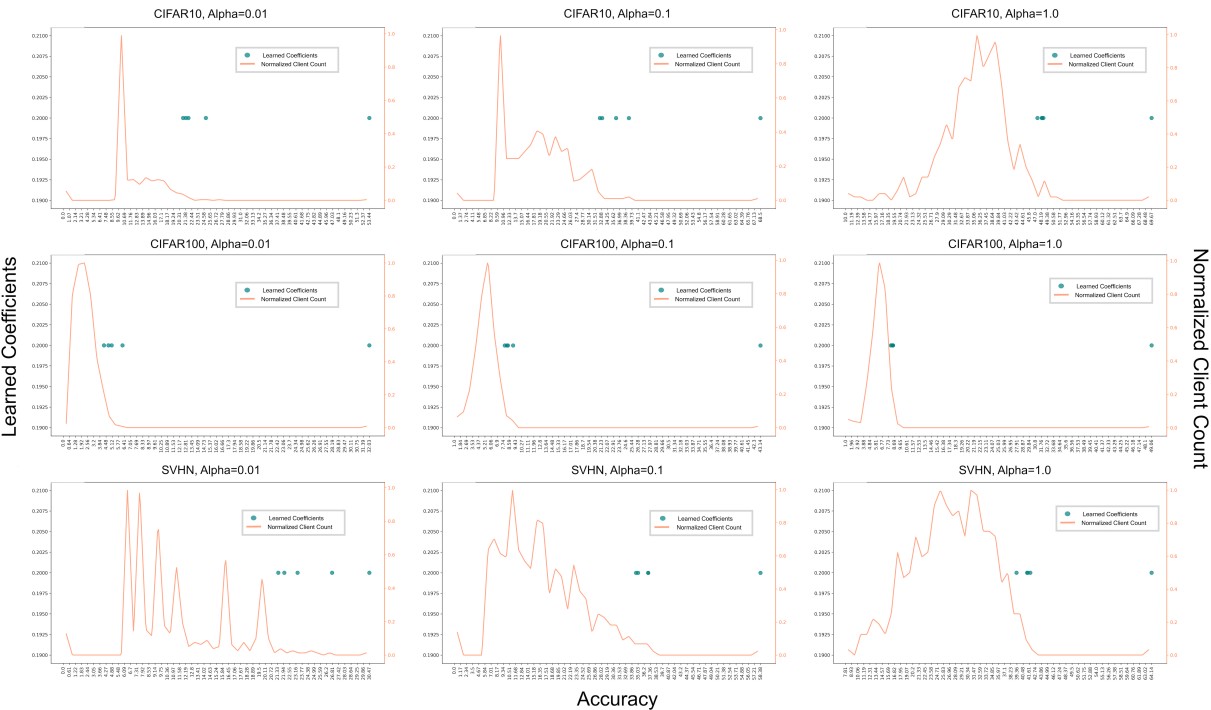

Figure 12: A sparsity analysis of the client selection process is presented. The green plot illustrates the correlation between client accuracy and their assigned selection weights, while the orange plot depicts the distribution of accuracies across clients, highlighting the five selected participants. The figure confirms the sparse nature of the selection, demonstrating that the chosen clients are predominantly clustered within the high-accuracy region of the dataset.

Figures 11 and 12 provides deeper insight into the sparsity-inducing behavior of the optimization framework by plotting the relationship between each client's selection weight and its corresponding target accuracy. The green scatter plot shows client accuracy (x-axis) against the learned selection weight assigned by FedSCM (y-axis). This visualization clearly demonstrates that only a small number of clients—those achieving relatively high accuracy—are assigned substantial weights, while the majority receive weights near zero. This reflects the sparsity guarantee of the underlying optimization formulation. Overlaid on this is an orange histogram, showing the overall distribution of client accuracies across the entire pool. Each panel in this figure is plotted on different vertical scales to better highlight trends specific to each dataset and $\alpha$ setting.

Two important conclusions emerge from this figure. First, the learned selection weights are highly sparse, confirming that FedSCM effectively identifies a small and relevant subset of clients, even when many clients are present. Second, the selected clients are concentrated in the high-accuracy tail of the distribution, indicating that the method consistently favors clients whose models generalize well to the target domain—despite having no access to labels or distribution statistics. This ability to isolate top-performing clients in a data-free manner is central to the success of source-free adaptation in heterogeneous federated settings.

All results correspond to a selection threshold of $\gamma = 0.7$, under which only 4 clients were ultimately selected. This strict threshold further underscores the discriminative power of the proposed method, allowing for extremely selective aggregation with minimal communication overhead while preserving or even improving downstream adaptation performance.

