# OpenReview forum: "FedSCM: Federated Domain Adaptation via Sparse Consensus Matching"
_TMLR — Under review for TMLR_

### Review · Reviewer_JX4j · 2026-06-10

**Summary Of Contributions:**

The paper studies the problem of selecting among a finite set of classifiers the ones better suited to classify unlabeled data, with only access to the classifiers themselves, and no other statistics on the training data than the size of the dataset.

To solve this problem, the paper relies on the following heuristic: classifiers that are more "confident" on the server dataset (i.e. with a low output entropy). Based on that, the paper introduces FedSCM, a method that selects the best convex combination of the original classifiers to classify the unlabeled data. The authors also introduce a regularized version of this algorithm that penalizes sources with smaller datasets, which plays the role of a penalization by the variance.

Finally, the paper studies the theoretical properties of the methods, presents optimization algorithms to solve the optimization problems useful to learn the weights of the method, and compares their method to other methods from the literature.

**Additional Comments:**

- In the abstract, it should be $n \geq \Omega (\log K)$. Written as it is, it is non-informative since, for instance, $0 \leq O(\log K)$.
- “This sparsity not only enhances computational efficiency.” I think it would be needed to say that this is true for future inference, as the sparsity of the solution does not seem to simplify the optimization problem, and the dataset has to be evaluated on each client’s hypothesis.

**Audience:**

Yes

**Audience Explanation:**

The paper perfectly fits in the scope of TMLR.

**Claims And Evidence:**

No

**Claims Explanation:**

Some of the results of the article are false under the stated assumptions, some oversell results, and some others are written under assumptions that appear abusive if not contextualized with a simple example of a problem satisfying them.

For instance:

- Theorem 3 cannot be true without extra assumptions. For instance, if all the classifiers from all the clients are the same, the optimization landscape is flat and any combination of lambdas yields the same value of the objective function and thus the solution is not always sparse. I think there needs to be a measure of complexity on the different hypotheses, like a shattering condition.

- Definition 4 looks extremely strong. For instance, if we look at the model for logistic models, it doesn’t satisfy it yet it is one of the simplest statistical learning problems. I do not think that this is a credible hypothesis, and I encourage the authors to present a simple example presenting a credible (but maybe toy) instance of the problem for which this definition applies in order to demonstrate that we are not simply reasoning on the empty set.

- For assumption 5, it is lacking some details. Here $H(P(\cdot | X))$ is a $X$-measurable random variable, so the assumption needs a qualifier on $X$, and I guess that what is missing is to say that it has to hold $X$-almost surely. But even with this fix, I do not think that this assumption can lead to meaningful upper-bounds in realistic scenarios. Indeed, $P(. | X)$ lives on the simplex of dimension number_of_classes - 1, and thus $H(P(. | X)) \leq \log(number_of_classes)$ by the usual upper bound saying that the entropy is maximal on the uniform distribution. So the assumption is only useful if it can be applied for some $b$s smaller than this trivial upper bounds, but it is not the case. Indeed, even on a simple non-degenerate logistic model, this upper bound is reached for some inputs.

- Without a simple example, it is impossible to tell if Theorem 6 is meaningful or not.

- Since the proof of Theorem 8 relies on the same arguments as the one of Theorem 3 and since Theorem 3 cannot hold in all generality for the reasons explained previously, Theorem 8 cannot hold in all generality written as it is.

- I do not understand the claim of ‘linear convergence’ in Theorem 8. The proof does not define any error metric satisfying a geometric contraction, and the argument seems instead to rely on monotone objective decrease plus sparsity/extreme-point structure, which would at best suggest a finite-termination type argument rather than linear convergence. Moreover, Algorithm 1 contains no line search, trust region, or merit-function acceptance step, so I do not see what mechanism guarantees that the true objective strictly decreases at each SLSQP iteration.

**Requested Changes:**

Critical:
- The basic heuristic underlying the method seems unsound under covariate shift. Consider a simple 2D logistic model trained on source data with Bayes separator x_2 = 0. If the target problem is the same model shifted to separator x_2 = a, then the source classifier becomes almost surely very confident on target inputs as a grows, simply because the sigmoid saturates far from 0, yet its target error tends to 1/2. Hence, low predictive entropy is not, by itself, a credible proxy for target-domain relevance. Can the authors comment on this and explain in the paper under what conditions on the distributions, the heuristic on which the paper is based is a good proxy for problem alignment?
- Can the authors comment on each of my previous concerns on the soundness of the paper?

Recommendations:
- By design (see definition 2 where the classifier is a linear combination of original classifiers), the considered classifiers are necessarily linear in the original ones. Can the authors provide a small discussion on whether this is restrictive?
- I would help the paper if the authors could present a principled way to choose the gamma.

---

> ### Comment · Action_Editor_hzJx · 2026-06-10
> **Issues with formatting**
>
> Dear reviewer,
> One of your items in the bullet list didn't format correctly, it looks like "$" is missing after an equation. Please update the review to fix this.

---

> > ### Comment · Reviewer_JX4j · 2026-06-10
> >
> > Dear action editor,
> >
> > Thank you, I’ve edited the review.

---

> ### Author Response · Authors · 2026-06-25
> **Part 1/3**
>
> We would like to thank the reviewer for their insightful comments and for their positive assessment of our manuscript. We appreciate the time and effort devoted to reviewing our work. Below, we provide detailed responses to the reviewer's comments.
>
> ---
>
> **Reviewer's comment**: (Theorem 3 and 8 validity) The reviewer notes that Theorem 3 cannot hold without extra assumptions (e.g., if all client classifiers are identical, the solution is not sparse). Since Theorem 8 relies on Theorem 3, it is also affected. A measure of complexity or non-degeneracy condition is suggested.
>
> **Our Response**:
> We thank the reviewer for this precise and valid observation. The counterexample is correct: when all client classifiers are identical, all $\lambda$ vectors within constraints become optimal. In fact, even if two models output the same probabilities for all samples, similar problems (to a lesser degree) can occur.
>
> **The Source of the Issue**: The objective in Definition 2 is concave, but the proofs of Theorems 3 and 8 implicitly relied on the stronger property of "strict" concavity (e.g., the proof of Theorem 8 assumes the Hessian is always negative-definite, i.e., $H^{(t)} \prec 0$). Some basic algebra reveals that strict concavity holds iff the clients' $nC$-dimensional prediction vectors $(h_k(X))_{X \in D}$ for $k\in[K]$ are linearly independent.
>
> **Proposed fix:** We propose a practical remedy: prior to running FedSCM, we discard linear dependencies among client prediction vectors. Concretely, we construct the $nC \times K$ matrix of client predictions over $\mathcal{D}$ and require it to be full column rank. If not, we greedily remove clients whose predictions lie in the span of others, retaining a maximal linearly independent subset.
>
> This is practically sensible: if client models are linearly dependent on server data, they are indistinguishable (at least via our objective), so keeping all of them serves no purpose. We will explicitly add this non-degeneracy condition to Theorems 3 and 8, as it is necessary and sufficient for strict concavity and our original sparsity and convergence conclusions. While we have not encountered degeneracy in practice, we will add a caveat in the revision: when models are *nearly* dependent, the Hessian remains negative-definite but may be ill-conditioned, which can potentially reduce the numerical stability of the sparsity pattern.
>
> ---
>
> **Reviewer's comment**: (Realism of Definition 4) The reviewer argues that Definition 4 is extremely strong and doesn't apply to common models like logistic regression. They request a credible toy example to demonstrate the hypothesis is not reasoning on an empty set, which would clarify if Theorem 6 is meaningful.
>
> **Our Response**:
> We thank the reviewer for raising this point; it allows us to clarify the motivation behind Definition 4, which we agree requires more explanation.
>
> **The modeling philosophy.** Definition 4 is our attempt to provide a mathematically tractable formulation of a phenomenon pervasive in federated learning under distribution shift: a classifier trained on a source domain, when evaluated on a target domain, behaves as if it partially tracks the true label distribution and partially produces uninformative or misleading outputs.
>
> When a learner has access to the true conditional $P(\cdot | \boldsymbol{X})$, the optimal model is the Bayes classifier. Under distribution shift or limited data, the learned model deviates from this ideal. A natural way to model this deviation is through a mixture:
> $$h(\boldsymbol{X}) = \eta P(\cdot | \boldsymbol{X}) + (1 - \eta) Q,$$
> where $Q$ represents the "derailment" from the Bayes classifier due to shift or noise, and $\eta \in [0,1]$ quantifies the client's expertise. When $\eta \to 1$, the client is essentially Bayes-optimal on the target; when $\eta \to 0$, the output is pure noise. This conceptual core captures how classifiers behave under distribution shift, regardless of architecture.
>
> **Why Dirichlet noise?** While $Q$ could be a simple uniform distribution, we deliberately chose $Q \sim \mathrm{Dirichlet}(\alpha, \ldots, \alpha)$ to make the model more general and challenging. A Dirichlet-distributed $Q$ can be **confidently wrong**—placing most of its mass on an incorrect label—rather than merely being uniformly uncertain. This ensures our guarantees hold even when irrelevant clients produce high-confidence but incorrect predictions. This is the regime where naive confidence-based selection fails and our consensus-alignment component becomes essential.
>
> **A concrete example.** Consider binary classification ($C = 2$) with a Naive Bayes classifier. Suppose the source and target share the same feature distribution $P_\mathcal{X}$ but differ in class-conditionals. The posterior output under such label shift takes the form $\eta P(\cdot|\boldsymbol{X}) + (1-\eta)Q$, where $Q$ depends on the source-target mismatch and $\eta$ is determined by the overlap between distributions.

---

> > ### Author Response · Authors · 2026-06-25
> > **Part 2/3**
> >
> > For appropriate choices of distributions, $Q$ follows a Dirichlet distribution, providing a concrete, non-vacuous instance of Definition 4.
> >
> > Definition 4 is an idealized model, much like Gaussian or linear models in statistical learning theory; it is intended to enable clean, illuminating analysis rather than literal representation. Crucially, we validate in Section 6 that the Dirichlet noise model is a good approximation for real neural networks on benchmark datasets, showing the model is not disconnected from practice.
> >
> > We added a detailed paragraph after Definition 4 explaining this modeling philosophy and make the assumption more transparent for the reader.
> >
> > ---
> >
> > **Reviewer's comment**: (Assumption 5 limitations) The reviewer points out that Assumption 5 needs an almost-sure qualifier on $X$. Even then, they argue it may not lead to meaningful bounds because the entropy bound $H(P(\cdot\vert X)\leq\log C$ is reached for inputs near the decision boundary in simple models.
> >
> > **Our Response**:
> > We thank the reviewer for mentioning this crucial weakness, and we acknowledge the problem the reviewer is pointing to: as originally stated, Assumption 5 lacks a qualifier on $X$, and even once stated almost surely, requiring $H(P(\cdot\vert X)) \leq b$ for \emph{every} (or almost every) $X$ is only meaningful if $b$ is strictly bounded away from $\log C$, which might fail to hold even for simple models.
> >
> > We propose to relax the assumption to either: i) hold on a large fraction, e.g, $1-\varrho$ for some small $\varrho$, of the $n$ input samples rather than throughout $\mathcal{D}$. Or ii) More technically, we can replace Assumption 5 with the requirement that
> > $$
> > P\Big( H(P(\cdot \vert X)) \leq b \Big) \geq 1 - \varrho,
> > $$
> > for some $b$ strictly smaller than $\log C$ and some small $\varrho \in [0,1)$.  Both i) and ii) lead to very similar bounds, however, i) is less technical and easier to work with. This explicitly allows a small fraction $\varrho$ of ``hard'' or near-boundary inputs to have entropy as large as $\log C$, while requiring the assumption to be non-trivial (i.e., $b < \log C$) on the remaining $(1-\varrho)$-fraction of the input distribution. We emphasize that this strict separation $b < \log C$ on the high-probability event is essential and is what restores the assumption's usefulness; without it, the relaxation alone would not address the reviewer's underlying concern.
> >
> > This relaxation has two consequences:
> > - introduces an additional parameter, $\varrho$, alongside the existing $(b, \delta, \delta')$.
> > - The sample-complexity condition in Theorem 7 is correspondingly modified to
> > $$
> > (1-\varrho)\frac{b+\delta}{(1-\delta')^2} \log\left(\frac{1}{b+\delta}\right)
> > +\frac{\varrho \log C}{(1-\delta')^2}
> > \leq \frac{\Psi_{C,\alpha}}{k^{*2}},
> > $$
> > so that the tuple $(\varrho, b, \delta, \delta')$ must now jointly be small for recovery guarantee.
> >
> > We have revised Assumption 5 to include the explicit almost-sure (or, in the relaxed version, high-probability) qualifier on $X$, add a footnote stating that $b$ must be strictly bounded away from $\log C$ for the assumption to be non-trivial, and add a dedicated appendix section deriving the refined sample-complexity condition above under the relaxed version of the assumption.
> >
> > ---
> >
> > **Reviewer's comment**: (Convergence Claims in Theorem 8) The reviewer questions the "linear convergence" claim, noting the lack of an error metric satisfying geometric contraction. They also observe that Algorithm 1 lacks mechanisms (like line search) to guarantee strict objective decrease at each SLSQP iteration.
> >
> > **Our Response**:
> > We thank the reviewer for this comment. We agree on both points. First, the term ``linear convergence'' is incorrect: our argument establishes no error metric exhibiting geometric contraction, and we will remove this claim from Theorem~8. Second, the reviewer is right that, as currently written, Algorithm 1 provides no per-iteration guarantee that the true objective strictly increases. Since the Hessian $H(\lambda)$ depends on $\lambda$ itself, without a line search, trust region, or merit-function acceptance criterion, a full step is not guaranteed to increase the true objective even when it improves the local quadratic model.
> >
> > **What we can establish instead.** What our argument does correctly support is the following: any limit point of the sequence  $\{\boldsymbol{\lambda}^{(t)}\}$ generated by Algorithm 1 satisfies the KKT conditions of the optimization problem in Definition 2. Under the non-degeneracy condition discussed in our response to the previous comment (full column rank of the
> > prediction matrix, ensuring strict concavity), any KKT point must be an extreme point of the feasible polytope, and hence has the sparsity structure guaranteed by Theorem 3. Since the feasible polytope has finitely many extreme points, the algorithm cannot cycle indefinitely among a continuum of non-sparse points: \emph{if} it converges, it converges to a sparse solution.

---

> > > ### Author Response · Authors · 2026-06-25
> > > **Part 3/3**
> > >
> > > We have revised Theorem 8 to state precisely this limit-point guarantee, rather than the stronger (and  unsupported) claim of monotone decrease at every iteration.
> > >
> > > We note that a genuine per-iteration monotone-decrease guarantee could be recovered by adding an Armijo-type step-size rule to  Algorithm 1: after computing the SLSQP step direction, one checks whether the true objective has sufficiently increased, and shrinks the step by a constant factor until it does. However, as this would add further technical machinery and lengthen the paper beyond what is necessary to convey the core contribution, we opt instead to tone down Theorem~8 to the limit-point statement above, which is both correct and sufficient for the purposes of the paper.
> > >
> > > ---
> > >
> > > **Reviewer's comment**: (Heuristic under Covariate Shift) The reviewer suggests the entropy heuristic may be unsound under covariate shift. For a logistic model, source classifiers can become very confident on shifted target inputs (due to sigmoid saturation) while having high error. They ask for clarification on when this heuristic is a good proxy for alignment.
> > >
> > > **Our Response**:
> > > We thank the reviewer for this illuminating example. We agree with the analysis: low predictive entropy alone is not a reliable proxy for relevance under all shifts. Based on our response to **Reviewer Egtd**, and due to NFL theorem, no selection heuristic is universal; a method optimized for label shift may be vulnerable to specific covariate shifts.
> > >
> > > **Label/Quantity Shift (Primary Regime):** FedSCM is designed for settings where the labeling concept $P(y|X)$ is shared, but marginals $P_i(y)$ differ. Here, low entropy genuinely reflects alignment between the client's concept and the target. Our Dirichlet model (Def. 4) and Theorems 3 & 8 formalize this, treating high confidence as a reliable signal of "expertise."
> > >
> > > **Covariate Shift & Robustness:** Under covariate shift, a model may be "confidently wrong" due to sigmoid saturation on out-of-distribution inputs. However, FedSCM does not rely on individual confidence alone. The **consensus-alignment** component (minimizing the entropy of the *combined* prediction) acts as a corrective filter. A client that is confidently wrong will likely disagree with relevant clients; this disagreement raises the mixture entropy, penalizing that client. While this provides robustness, it is not an absolute safeguard if most clients share the same shift bias.
> > >
> > > **Proposed Revision:** we explicitly characterized these regimes in the theory section and a new limitations paragraph. We will clarify that while consensus provides partial robustness against saturating classifiers under covariate shift, the entropy heuristic is most reliable under shared-concept shifts (label/quantity).
> > >
> > > ---
> > >
> > > **Reviewer's comment**: (Restrictiveness and Efficiency) The reviewer asks if the linear combination of classifiers is restrictive and notes that the sparsity benefits "future inference," as the optimization itself still requires evaluating all client hypotheses.
> > >
> > > **Our Response**:
> > > We thank the reviewer for the point. We wish to clarify a misunderstanding of Definition 2: the convex combination $\sum_i \lambda_i h_i(X)$ is **not** the final inference model. It is purely an auxiliary construction for the **selection stage**.
> > >
> > > The weights $\lambda_i$ act as soft selection flags; their induced mixture entropy is simply the objective function. Once optimization is complete, we only retain the **support** of optimal $\lambda$ (clients where $\lambda_i^* > 0$). The specific weight values are discarded (see Section 5). The actual downstream model is produced by running a standard federated procedure (e.g., FedProx) exclusively on those selected clients. This fine-tuning stage is independent of the linear combination used during selection. Consequently, the final model's expressivity is determined by the shared architecture and the fine-tuning process, not by the selection construction.
> > >
> > > ---
> > >
> > > **Reviewer's comment**:
> > > **Choosing Hyperparameters**: The reviewer suggests the paper would benefit from a principled way to choose $\gamma$.
> > >
> > > **Our Response**:  We have already responded to a similar comment from Reviewer bUVF. We refer the reviewer to that discussion.
> > >
> > > ---
> > >
> > > **Minor comment on** $n\ge O(\log K)$:
> > > The reviewer is correct. We will replace $\mathcal{O}(\log K)$ with $\Omega(\log K)$.

---

> > > > ### Comment · Reviewer_JX4j · 2026-07-07
> > > >
> > > > I thank the authors for their answer. I still have a few remarks.
> > > >
> > > > - Regarding the sparsity claim, I think that the rank constraint that is added solves the problem. However I think that it adds the constraint that $n \gtrsim K$. If so, the lower-bound on $n$ has to be updated.
> > > >
> > > > - Regarding the linear convergence claim, I thank the authors for updating their theorem. However, such a claim still appears within the text on page 4.
> > > >
> > > > - Regarding the strong assumptions, I am not certain, even after the rebuttal, to be convinced that they are realistic. However, since the other reviewers accept them, I will consider this issue resolved.

---

> > > > > ### Author Response · Authors · 2026-07-15
> > > > >
> > > > > We would like to thank the reviewer for their second-round comments. We have revised the manuscript and uploaded a new version. The two remaining issues raised in the review have now been addressed, and the relevant claims and theorems have been revised accordingly.
> > > > >
> > > > > Please let us know if any further revisions are required.
> > > > >
> > > > > Thank you for your time and consideration.

---

### Review · Reviewer_Egtd · 2026-06-15

**Summary Of Contributions:**

## Summary
The paper addresses model/client selection in a federated domain adaptation setting with heterogeneous sources, where the server holds only unlabeled target data and has access to client models but not their training data. The proposed method performs entropy minimization over client weights, selecting those with non-zero weights as relevant clients. The number of selected clients depends on a hyperparameter. Theoretical results establish that the selected set of clients is sparse by construction and that under some assumptions on the relatedness among the client and server data distributions, the selection is optimal when the number of target samples is at least $O(\log K)$, where $K$ is the number of clients. Empirically, the selected clients achieve high accuracy on the target distribution and the method outperforms existing approaches in this specific setting.

## Strengths
1. The paper addresses a practical problem: how to identify useful client models when the server has access only to unlabeled target data and the models.
2. The empirical evaluation includes an extensive comparison to prior work.
3. The paper is in general carefully written with attention to detail.

## Weaknesses:
1. The privacy motivation is insufficiently justified. The authors argue that operating on models rather than raw data preserves privacy. However, it is well established that models can leak sensitive information about training data (for example through membership inference attacks). This claim should be revised or softened in the final version of the paper.
2. There are minor issues in the text that can be fixed and some points that would benefit from further clarification (see detailed comments below).

**Additional Comments:**

Based on the method description in Definition 2 and the experimental results (Tables 2,3,4), it seems that FedSCM is better suited for label and quantity shift than for covariate shift. I would be interested in the author's opinion on this.

**Audience:**

Yes

**Audience Explanation:**

The paper will be interesting to a broad audience working in federated learning, domain adaptation, metalearning and related areas.

**Broader Impact Concerns:**

No concerns

**Claims And Evidence:**

Yes

**Claims Explanation:**

Overall, the claims are supported by accurate, convincing and clear evidence. The theorems are clearly stated and the proofs appear correct. The assumptions underlying the theoretical analysis are strong but reasonable given the setting. The experiments support the main promises of the method.

Two claims, however, require attention:
1. **Privacy**: The claim that the method preserves privacy is not adequately supported. While it is plausible that the method could be extended to incorporate privacy-preserving mechanisms the current paper does not do so. The claims in the abstract and the introduction should be revised to reflect the current method.
2. **Communication and computation overhead**: The claim that the method reduces communication and computation overhead would benefit from more precise support. If such statements already appear in the paper, the authors should make them more prominent, otherwise additional clarification is needed.

**Requested Changes:**

1. **Privacy claim**: Revise the claims in the abstract and introduction to accurately reflect what the method provides.
2. **Communication and computation overhead**: Add more precise clarification to support this claim.
3. **Minor issues**:
- The label shift definition on page 13 states that any two clients $i,j$ share the same conditional distribution $P_i(y\mid X) = P_j(y \mid X)$. This appears to be incorrect. Under label shift, it is typically the marginal distributions that differ $P_i(y)\neq P_j(y)$, while the class-conditional distributions agree $P_i(X\mid y) \neq P_j(X\mid y)$.
- Can you elaborate on the sampling procedure used to introduce label and quantity shift in the experiments?
- In Section 6.3.5, can you elaborate on the estimation procedure, particularly what is used as $P(\cdot\mid X)$?

---

> ### Author Response · Authors · 2026-06-25
>
> We would like to thank the reviewer for their insightful comments and for their positive assessment of our manuscript. We appreciate the time and effort devoted to reviewing our work. Below, we provide detailed responses to the reviewer's comments.
>
> ---
>
> **Reviewer's comment**: The privacy motivation and privacy-related claims should be softened and better justified.
>
> **Our Response**: We thank the reviewer for this important observation, and we agree. Sharing trained models rather than raw data does not constitute a formal privacy guarantee, as models remain vulnerable to attacks such as membership inference, model inversion, and parameter leakage. Our current wording conflates two distinct notions: (i) the standard FL practice of not transmitting raw data, and (ii) formal privacy guarantees such as differential privacy, which our method does not provide.
>
> **Proposed Revision:** We have revised the introduction to clarify that the privacy benefits of $\mathsf{FedSCM}$ are informal and stem only from avoiding raw-data sharing. We explicitly stated that the method does not provide formal privacy guarantees and that integrating mechanisms such as differential privacy is left to future work.
>
> ---
>
> **Reviewer's comment**: The communication and computation overhead claims should be supported more explicitly.
>
> **Our Response**: We thank the reviewer for this comment. The reduction in communication and computation overhead follows directly from the sparsity guarantee of Theorem 3: since $\mathsf{FedSCM}$ selects at most $\lceil 1/(1-\gamma) \rceil$ clients out of $K$ for the subsequent federated fine-tuning stage, the associated communication and computation cost is reduced by a factor of $K / \lceil 1/(1-\gamma) \rceil$ relative to full participation. We agree that this connection is currently implicit.
>
> Please note that no other claims regarding the reduction of communication/computation overhead, except the case described above, have been made; if any others are found, we will remove them.
>
> **Proposed Revision:** We explicitly quantified this reduction in the discussion following Theorem 3 and make the resulting savings more prominent throughout the paper.
>
> ---
>
> **Reviewer's comment**: Clarification regarding the definition of label shift.
>
> **Our Response:** We thank the reviewer for the careful reading. Our intention was to describe the setting in terms of a shared labeling concept, expressed as $P_i(y\vert X)=P_j(y\vert X)$, together with differing label marginals $P_i(y)\neq P_j(y)$. We agree, however, that this differs from the standard label-shift formulation typically stated as $P_i(X\vert y)=P_j(X\vert y)$ with varying class priors. Because our discussion focused only on $P(y\vert X)$, the terminology may have been confusing and insufficiently connected to the standard domain-adaptation literature.
>
> **Proposed Revision:** We have revised the discussion to explicitly distinguish our assumption from the classical label-shift assumption, clarify the relationship between the two formulations via Bayes' rule, and ensure that the terminology used throughout the manuscript is consistent. We will also review the remainder of the paper (including the appendix) to eliminate any ambiguity.
>
> ---
>
> **Reviewer's comment**: Please elaborate on the procedure used to generate label and quantity shift in the experiments.
>
> **Our Response**: We thank the reviewer for this question. We follow the standard federated-learning protocol based on Dirichlet sampling. For each client, a class-proportion vector is sampled from $\mathrm{Dirichlet}(\alpha,\ldots,\alpha)$ with $\alpha\in{1,0.1,0.01}$, and local datasets are constructed according to these proportions. Smaller values of $\alpha$ induce stronger heterogeneity and more severe class imbalance. Because the number of available examples per class is finite, this procedure also naturally induces quantity shift, as some clients receive substantially more samples than others.
>
> ---
>
> **Reviewer's comment**: Clarification regarding the estimation of $\hat{\eta}$ in Section 6.3.5.
>
> **Our Response**: The estimate $\hat\eta_k$ is obtained by directly fitting the Dirichlet noise model of Definition 4 to the client's actual predictive outputs, rather than via any proxy based on label agreement or accuracy. Specifically, for each candidate value $\eta \in [0,1]$, we ask how well the mixture $\eta P(\cdot\vert X) + (1-\eta)Q_{X}$ (with $Q_{X}$ Dirichlet-distributed) explains client $k$'s observed predictive distribution $h_k(X)$, measured in KL-divergence averaged over $\mathcal{D}$, and $\hat\eta_k$ is the minimizer of this divergence. In other words, $\hat\eta_k$ is the best-fit mixture weight under the assumed generative model, found by direct optimization against the client's own outputs. We believe this is close to the canonical way to estimate $\eta$ under our model, and expect other reasonable estimators targeting the same fitting objective to yield similar values.

---

> > ### Author Response · Authors · 2026-06-25
> >
> > **Reviewer's comment**: FedSCM appears better suited for label and quantity shift than for covariate shift.
> >
> > **Our Response:** We thank the reviewer for this insightful observation, and we agree with this assessment. This is, in fact, a structural property of the entropy-based selection mechanism underlying $\mathsf{FedSCM}$, which a different reviewer also raised in relation to the validity of the entropy heuristic under covariate shift; we discuss this in detail in our response to that comment, and have accordingly added a dedicated discussion to the paper (end of Section 4.2).
> >
> > In brief: under label and quantity shift, the labeling concept $P(y\vert X)$ is shared across clients and the server by definition, so low predictive entropy on $\mathcal{D}$ is a reliable proxy for relevance. Under covariate shift, the input distribution itself differs, and a client's model may become confidently wrong on target inputs far from its source training distribution due to classifier saturation, making individual confidence a less reliable signal. The consensus-alignment term in the $\mathsf{FedSCM}$ objective partially mitigates this, since a confidently-wrong client due to covariate shift is unlikely to agree with other relevant clients --- consistent with $\mathsf{FedSCM}$ still outperforming baselines under covariate shift, though by a smaller margin than under label/quantity shift, as observed in Tables 2--4.

---

### Review · Reviewer_bUVF · 2026-06-21

**Summary Of Contributions:**

The authors propose FedSCM, a client selection method in federated domain adaptation. The problem statement addressed is that the server has access to an unlabeled target dataset and has to choose from a set of clients based on their right data distributions that aligns with the target data. The key idea is to choose clients such as the entropy of the weighted mixture of predictions is minimized.

Strengths:
1. This is an important problem in the space of federated learning, and the authors propose a neat framework backed by theoretical proofs that works.The problem formulation is clear and well-motivated.
2. The experimental design is strong. The authors validate their findings via 3 datasets, 500 clients, multiple runs with correct baselines.

Weakness:
Overall, I really like the paper, so I dont want to call this as a strong weakness per say. But the hyperparameter selection method is not clear in the paper (γ). How is the this value fixed without labeled target data. If the cross-validation step requires some ground-truth labels in the server, that would violate some assumptions in the problem statement, i.e server is target labels unaware.

**Audience:**

Yes

**Audience Explanation:**

Yes, this paper solves an important problem in the space of federated learning.

**Broader Impact Concerns:**

No ethical concerns.

**Claims And Evidence:**

Yes

**Claims Explanation:**

The experimental results shown in the paper are strong and is conducted against 3 datasets, 500 clients showing the strengths of the proposed method.

**Requested Changes:**

Details on hyperparameter selection has to be provided to understand how the selection happens in the absence of target labels on the server side.

---

> ### Author Response · Authors · 2026-06-25
>
> We would like to thank the reviewer for their insightful comments and for their positive assessment of our manuscript. We appreciate the time and effort devoted to reviewing our work. Below, we provide detailed responses to the reviewer's comments.
>
> --------
>
> **Reviewer's comment**: The hyperparameter selection method is not clear in the paper ($\gamma$). How is the this value fixed without labeled target data. If the cross-validation step requires some ground-truth labels in the server, that would violate some assumptions in the problem statement, i.e server is target labels unaware.
>
>
> **Our Response**: We thank the reviewer for this insightful observation. According to Theorem 3, $\mathsf{FedSCM}$ selects at most $\lceil 1/(1-\gamma) \rceil$ clients out of $K$ for the subsequent federated fine-tuning stage. In practice, this result alone can guide the choice of $\gamma$: the server may have communication or computation limitations, or a capped incentive budget for inviting clients to participate in post-selection FL, and those constraints directly determine how many clients can be accommodated, hence the value of $\gamma = 1 - 1/s$ for a desired maximum of $s$ clients. Crucially, this requires no labeled target data whatsoever.
>
> A complementary, more data-driven approach is to inspect the value of the $\mathsf{FedSCM}$ objective in Definition 2 as a function of $\gamma$, and identify a point of diminishing returns (an ``elbow'') as $\gamma$ increases. This also requires no labeled data on the server side, as the objective is computed entirely on the unlabeled $\mathcal{D}$. Additionally, Theorem 7 establishes that, under its stated conditions, $\gamma$ can in principle be uniquely identified without labeled data; however, we acknowledge that the exact satisfaction of those conditions is not always verifiable in
> practice. In our own experiments, the primary criterion for choosing $\gamma$ was the desired number of participating clients.
>
> **Proposed Revision**: We have added a short paragraph to the experiments section clarifying how $\gamma$ was selected in our experiments and summarizing the practical guidance above, to make the hyperparameter selection procedure transparent to the reader.